# Concomitant gain and loss of function pathomechanisms in C9ORF72 amyotrophic lateral sclerosis

Arun Pal[1], Benedikt Kretner[1,2], Masin Abo-Rady[2], Hannes Glaβ[3], Banaja P Dash[3], Marcel Naumann[3], Julia Japtok[1], Nicole Kreiter[1], Ashutosh Dhingra[6], Peter Heutink[6,7], Tobias M Böckers[5], René Günther[1,8], Jared Sterneckert[2], Andreas Hermann[3,4]

Intronic hexanucleotide repeat expansions (HREs) in *C9ORF72* are the most frequent genetic cause of amyotrophic lateral sclerosis, a devastating, incurable motoneuron (MN) disease. The mechanism by which HREs trigger pathogenesis remains elusive. The discovery of repeat-associated non-ATG (RAN) translation of dipeptide repeat proteins (DPRs) from HREs along with reduced exonic C9ORF72 expression suggests gain of toxic functions (GOFs) through DPRs versus loss of C9ORF72 functions (LOFs). Through multiparametric high-content (HC) live profiling in spinal MNs from induced pluripotent stem cells and comparison to mutant FUS and TDP43, we show that HRE *C9ORF72* caused a distinct, later spatiotemporal appearance of mainly proximal axonal organelle motility deficits concomitant to augmented DNA double-strand breaks (DSBs), RNA foci, DPRs, and apoptosis. We show that both GOFs and LOFs were necessary to yield the overall *C9ORF72* pathology. Increased RNA foci and DPRs concurred with onset of axon trafficking defects, DSBs, and cell death, although DSB induction itself did not phenocopy C9ORF72 mutants. Interestingly, the majority of LOF-specific DEGs were shared with HRE-mediated GOF DEGs. Finally, C9ORF72 LOF was sufficient—albeit to a smaller extent—to induce premature distal axonal trafficking deficits and increased DSBs.

## Introduction

Amyotrophic lateral sclerosis (ALS, Table 1) is a devastating, incurable motoneuron (MN) disease. Hallmarks of ALS pathology are degeneration of spinal and cortical MNs causing progressive muscular paralysis, leading to death within 2–5 yr after the onset of clinical manifestation (Rothstein, 2009). MN degeneration progresses by retraction and dying back of axons from neuromuscular junctions to final death of somata (Frey et al, 2000; Fischer et al, 2004; Dadon-Nachum et al, 2011). We have previously modeled retrograde axonal dying back in vitro in a human cell model using compartmentalized induced pluripotent stem cell (iPSC)–derived spinal MNs from ALS patients (Naumann et al, 2018). A better understanding of the underlying pathomechanism is hampered by the multitude of genetic causes in familial and sporadic ALS. To date, more than 30 distinct mutations have been identified (Chia et al, 2018; Nguyen et al, 2018), ranging from single amino residue substitutions to truncations and intronic hexanucleotide repeat expansions (HREs). This diversity of affected genes and mutation types seems to contradict the common scheme of MN degeneration and the final clinical outcome in ALS and calls for a thorough, comprehensive dissection in clinically relevant models to reveal mutation-specific upstream versus more common downstream mechanisms during the progression of neurodegeneration. To this end, we are using fast multichannel live imaging on compartmentalized axons in vitro at standardized distal versus proximal readout sites (Naumann et al, 2018) owing to the hotly debated role of membrane trafficking defects in many neurodegenerative diseases (Sheetz et al, 1998; Salinas et al, 2008; Veleri et al, 2018). Using this setup, we have previously reported about deficient mitochondrial and lysosomal organelle trafficking in iPSC-derived spinal motor neurons from ALS patients bearing mutant fused in sarcoma (FUS) (Naumann et al, 2018) and TDP43 (Kreiter et al, 2018), two frequent genetic causes of ALS.

HREs of GGGGCC in intron 1 of *C9ORF72* are the most frequent causes of ALS, accounting for 40% of familial and 5% of sporadic cases (Majounie et al, 2012). HRE in *C9ORF72* is also one of the main genetic causes of frontotemporal dementia (FTD) (Suh et al, 2015; Van Mossevelde et al, 2017). GGGGCC repeat numbers range from 2 to 23 in healthy persons and are increased to at least 60 in ALS patients and beyond 1,000 in extreme cases (Suh et al, 2015; Van

[1]Division of Neurodegenerative Diseases, Department of Neurology, Technische Universität Dresden, Dresden, Germany    [2]Center for Regenerative Therapies TU Dresden (CRTD), Technische Universität Dresden, Dresden, Germany    [3]Translational Neurodegeneration Section "Albrecht-Kossel," Department of Neurology, and Center for Transdisciplinary Neuroscience (CTNR), University Medical Center Rostock, University of Rostock, Rostock, Germany    [4]German Center for Neurodegenerative Diseases (DZNE) Rostock/Greifswald, Rostock, Germany    [5]Institute for Anatomy and Cell Biology, Ulm University, Ulm, Germany    [6]German Center for Neurodegenerative Diseases (DZNE), Genome Biology of Neurodegenerative Diseases, Tübingen, Germany    [7]Department for Neurodegenerative Diseases, Hertie Institute for Clinical Brain Research, University of Tübingen, Tübingen, Germany    [8]German Center for Neurodegenerative Diseases (DZNE) Dresden, Dresden, Germany

Correspondence: Andreas.Hermann@med.uni-rostock.de

Mossevelde et al, 2017). *C9ORF72* is inherited in an autosomal dominant fashion, and the mechanism by which HREs cause ALS remains unclear. Because HREs concomitantly occur with reduced expression of the exonic *C9ORF72* gene (Waite et al, 2014; Sivadasan et al, 2016; Frick et al, 2018), a loss of its reported function (LOF) in axonal trafficking due to haploinsufficiency appeared feasible (Farg et al, 2014; Gendron & Petrucelli, 2018; Shi et al, 2018). However, various KO zebra fish and mouse models of *C9ORF72* failed to recapitulate MN degeneration and ALS pathology (Hruscha et al, 2013; Jiang et al, 2016; O'Rourke et al, 2016). Subsequently, the discovery of noncanonical RAN translation of neurotoxic dipeptide repeat proteins (DPRs) from intronic HREs (Zu et al, 2011; Cleary & Ranum, 2013) led to the hypothesis of a DPR-mediated GOF (Jiang et al, 2016). Specifically, DPRs are translated bidirectionally from both the sense and antisense HRE-RNA transcripts, resulting in a whole spectrum of DPR variants, among them are more abundant poly glycine–alanine (GA) and poly glycine–proline (GP) (Walker et al, 2017; Nihei et al, 2020). DPR expression confers its toxic GOF presumably through the formation of inclusion bodies that sequester phosphorylated ataxia telangiectasia mutated protein (pATM), a key player of DNA damage response, and heterogeneous ribonucleoprotein (hnRNP) A3, normally limiting DPR expression (Walker et al, 2017; Nihei et al, 2020). The resultant DNA damage accumulation eventually leads to neurodegeneration. Other HRE-mediated GOF mechanisms in concert comprise the formation of sense and antisense RNA repeat-expansion (RRE) foci resulting from bidirectional transcription (Walker et al, 2017). RRE foci confer RNA toxicity through erratic RNA processing and splicing. Moreover, during the transcription of HREs, nascent RNA is prone to hybridize with the DNA template strand, thereby displacing the complementary DNA strand and forming a three-stranded structure termed R-loops (Groh & Gromak, 2014; Walker et al, 2017), which directly increase the risk of DNA double-strand breaks (DSBs). But again, some mouse models testing GOF by overexpressing HREs and DPRs failed to recapitulate MN degeneration, particularly dying back events and ALS pathology (Chew et al, 2015; O'Rourke et al, 2015), whereas a recent novel transgenic mouse model expressing more toxic poly-PR showed at least some loss of spinal MNs (Hao et al, 2019), suggesting that DPR composition and expression technicalities of mouse models matter.

We have recently established isogenic lines of iPSC-derived spinal MNs comprising parental *C9ORF72* from ALS patients along with a (i) gene-corrected (GC) variant with intronic HREs excised, (ii) a KO of the exonic *C9ORF72* part with intronic HREs maintained, and (iii) a similar KO of *C9ORF72* in control cells with naturally no HREs (Abo-Rady et al, 2020). In this report, we use these GC and KO variants to dissect GOF from LOF in C9ORF72 pathology through high-content (HC) phenotypic live profiling of mitochondrial and lysosomal organelle trafficking in MN axons. This approach appeared particularly attractive in light of the documented roles of C9ORF72 in membrane trafficking (Farg et al, 2014; Sivadasan et al, 2016; Shi et al, 2018). Deficient trafficking in aged C9ORF72 MNs from patients was mirrored by DNA damage and DPR accumulation along with apoptosis. LOF of exonic C9ORF72 in the KO variant was further exacerbating perturbed trafficking and apoptosis because of the remaining HRE-mediated GOF, whereas the GC variant with no HREs was fully rescuing all phenotypes. Surprisingly, the "pure" LOF of

exonic C9ORF72 in the KO variant of control cells with naturally no HREs partially mimicked the trafficking, DNA damage, and apoptosis phenotype, thereby arguing against a sole role of HRE-mediated GOF.

# Results

## Live imaging of compartmentalized MNs revealed distinct organelle trafficking defects in C9ORF72 compared with FUS and TDP43

C9ORF72 has reported roles in endosomal and autophagic membrane trafficking (Farg et al, 2014; Sivadasan et al, 2016; Shi et al, 2018). Furthermore, MNs with HREs in *C9ORF72* showed decreased lysosomal axonal trafficking compared with GC MNs (Abo-Rady et al, 2020). Thus, we first wanted to compare trafficking deficits in *C9ORF72* lines with other typical ALS causing genes, that is, *FUS* and *TDP43*.

We selected a gender mix of five different ALS patients with confirmed heterozygous HREs in intron 1 of the *C9ORF72* gene locus with repeat numbers between 50 and 1,800 (C9-1, C9-2, C9-3, C9-4, and C9; Table 2), respectively, and compared them against three healthy control donors (Ctrl1, Ctrl2, and Ctrl3; Table 2). These lines were fully characterized and validated in previous publications (Table 2). Furthermore, we included our recently published phenotypic profiles from mutant *TDP43* and *FUS* (Pal et al, 2018) to compare them against HRE *C9ORF72*. All iPSC lines were matured to spinal MNs in microfluidic chambers (MFCs), in which only axons could reach and fully penetrate the microgroove barrier of channels from the proximal soma seeding site to distal exits (Naumann et al, 2018) (Fig 1A), thereby enabling axon-specific studies with defined antero- versus retrograde orientation. Of note, our differentiation protocol combined with 900 $\mu$m length of microgrooves resulted in exclusive penetration by MN axons, as we documented previously (Pal et al, 2018; Glaß et al, 2020). We performed fast dual-color live imaging of mitochondria and lysosomes at strictly standardized distal versus proximal readout positions as described (Naumann et al, 2018) on day D21. All movies were analyzed with FIJI TrackMate plugin to deduce organelle tracks with respect to mean speed and track displacement, the latter serving as measure for directed, processive movements as opposed to random walks. Our movie analysis was previously established to reveal distal axonal trafficking defects in mutant TDP43- and FUS-ALS (Kreiter et al, 2018; Naumann et al, 2018). Maximum intensity projections of entire movie stacks enabled a preliminary visual inspection for major alterations in motility patterns (Fig 2A and Videos 1 and 2). Directed, processive trafficking events were highlighted as long trajectories, whereas stationary organelles and nonprocessive "jitter" remained virtually as punctae. We obtained HC phenotypic signatures for each line, as recently described (Pal et al, 2018). In brief, each parameter was expressed as Z-score deviation from pooled control lines at the proximal readout and plotted with a connecting line to obtain the signature (Pal et al, 2018) (Fig 1B). A master set of 11 parameters was obtained four times owing to two readout positions (distal versus proximal) and two markers (Mito- and LysoTracker),

**Table 1.  Overview cell line characteristics.**

| Original name | Alias | Mutation | Exonic genotype | Intronic HRE genotype | Source | Gender | Year of birth | Age at biopsy (years) | Reference (first published) |
|---|---|---|---|---|---|---|---|---|---|
| T12.9 | Ctrl1 | Control, parental to KO line below | WT/WT | WT/WT | Dresden (Sterneckert) | F | 1959 | N/A | Reinhardt et al (2013) |
| T12.9 KO | WT-KO | KO of exonic *C9ORF72* in T12.9 (isogenic) | –/– | WT/WT | Dresden (Sterneckert) | F | 1959 | N/A | Abo-Rady et al (2020) |
| AKC5 | Ctrl2 | Control | WT/WT | WT/WT | Dresden (Hermann) | F | 1963 | 48 | Reinhardt et al (2013) |
| 30.1 | Ctrl3 | Control | WT/WT | WT/WT | Dresden (Sterneckert) | M | 1971 | N/A | Reinhardt et al (2013) |
| Pooled Ctrls | Ctrl | Pooled T12.9, AKC5, 30.1 | WT/WT | WT/WT | N/A | N/A | N/A | N/A | N/A |
| FUS-WT-EGFP C4 | FUS GC | WT FUS-EGFP (isogenic) | *FUS*-WT-EGFP/WT | WT/WT | Dresden (Hermann) | F | 1952 | 58 | Naumann et al (2018) |
| FUS-P525L-EGFP C21 | FUS | P525L FUS-EGFP (isogenic) | *FUS*-P525L-EGFP/WT | WT/WT | Dresden (Hermann) | F | 1952 | 58 | Naumann et al (2018) |
| TDP43 S393L | N/A | TDP43 S393L | *TDP43*/WT | WT/WT | Dresden (Hermann) | F | ND | 87 | Kreiter et al (2018) |
| TDP43 G294V | N/A | TDP43 G294V | *TDP43*/WT | WT/WT | Dresden (Hermann) | M | ND | 46 | Kreiter et al (2018) |
| Pooled TDP43 | TDP43 | Pooled TDP43 S393L and G294V | *TDP43*/WT | WT/WT | N/A | N/A | N/A | N/A | N/A |
| KDC28 | C9-1 | C9ORF-HRE (>50 rep.) | WT/WT | *C9ORF*-HRE/WT | Dresden (Hermann) | F | 1944 | 68 | Sivadasan et al (2016) |
| MHC30 | C9-2 | C9ORF-HRE (ca. 730 rep.) | WT/WT | *C9ORF*-HRE/WT | Dresden (Hermann) | F | 1961 | 51 | Sivadasan et al (2016) |
| 34.1 | C9-3 | C9ORF-HRE (>850) | WT/WT | *C9ORF*-HRE/WT | Dresden (Sterneckert) | | | | Donnelly et al (2013) |
| JBR | C9-4 | C9ORF-HRE (ca. 1800 rep.) | WT/WT | *C9ORF*-HRE/WT | Ulm (Böckers, Ludolph, Demestre) | M | 1951 | NA | Higelin et al (2018), Catanese et al (2019) |
| 33.1 | C9 | C9ORF-HRE (>620 rep.), parental to isogenic lines below | WT/WT | *C9ORF*-HRE/WT | Dresden (Sterneckert) | M | 1944 | 65 | Donnelly et al (2013) |
| 33.1 GC | C9-GC | Gene-corrected 33.1 (isogenic) | WT/WT | WT/WT | Dresden (Sterneckert) | M | 1944 | 65 | Abo-Rady et al (2020) |
| 33.1 KO | C9-KO | KO of exonic *C9ORF72* in 33.1 with preserved intronic HREs (isogenic) | –/– | *C9ORF*-HRE/WT | Dresden (Sterneckert) | M | 1944 | 65 | Abo-Rady et al (2020) |
| Pooled C9ORFs | C9ORF | Pooled KDC28, JBR, MHC30, 34.1, 33.1 | WT/WT | *C9ORF*-HRE/WT | N/A | N/A | N/A | N/A | N/A |

assembled to a signature of 44 parameters in total (Pal et al, 2018). Only Z-scores below −5 and above 5 were considered significant deviations because of established conventions (Pal et al, 2018) (grey horizontal lines, Fig 1B). As all HRE *C9ORF72* lines showed very similar signatures on D21 (in pink, Fig S2), we averaged their data to obtain a pooled profile (C9ORF, in red, Fig 1B). In essence, C9ORF displayed a flat line similar to controls on D21 (red versus blue, Fig 1B), consistent with no phenotype in the raw data (Figs 2A and S1 and Videos 1 and 2), in track displacement and mean speed (Fig 2C and Videos 1 and 2). The FUS (in grey) and TDP43 (in green) signatures (Fig 1B) were

virtually identical to our recent report (Pal et al, 2018), except they were normalized to the pooled control lines used in this study (Ctrl, Table 2) and not to the isogenic *FUS* GC line (Pal et al, 2018). Again, mutant FUS and TDP43 showed pronounced reductions in many parameters, which were particularly pronounced in distal FUS axons, essentially indicating a distal axonopathy in both TDP43 and FUS.

Although the Z-scores indicated to what extend a single parameter deviated from control conditions, we strived to have an objective measure of comparing entire multiparametric signatures and grouped them based on similar phenotypic traits. We

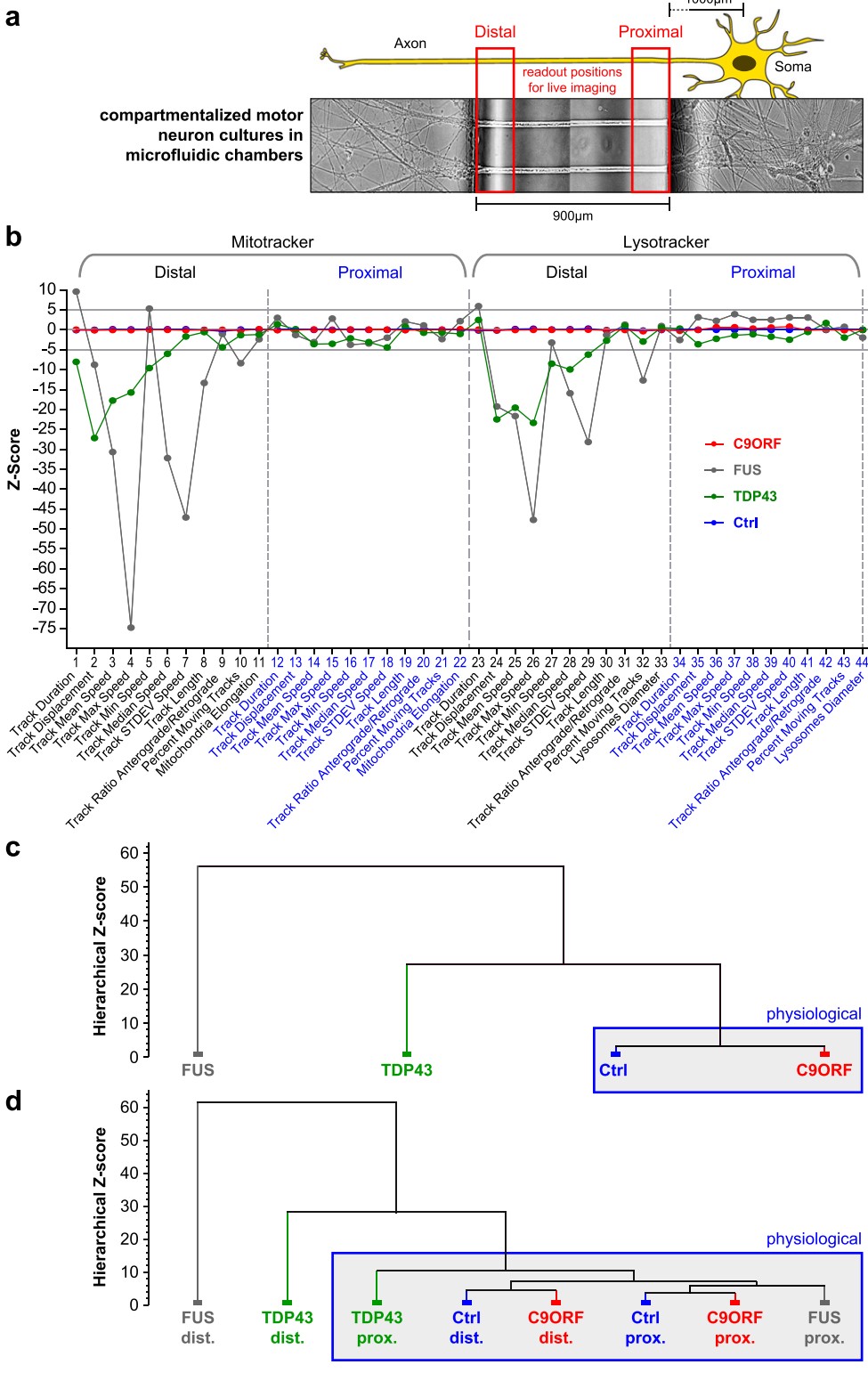

**Figure 1. Multiparametric high-content phenotypic profiling revealed no phenotype in C9ORF72 at D21 of spinal MN maturation.**
**(A)** Schematic live setup of motor neurons (MNs) in zona microfluidic chambers (MFCs). The central microgroove of channels formed a physical barrier between the distal (left) and proximal (right) site where the somata were seeded. Only axons, not dendrites, could penetrate the microchannels. **(B)** Multiparametric high-content signatures corresponding to the maximum intensity projections in Fig 2A, D21. Shown is the Z-score deviation of each tracking parameter from the proximal readout of pooled Ctrl lines (in blue). A set of 11 parameters (bottom labels) was deduced for both the Mito- and LysoTracker, distal versus proximal each, as indicated in the header, resulting in 44 parameters in total. The signatures of mutant FUS (in grey) and TDP43 (in green) were taken from our previous publication to facilitate the comparison to C9ORF (in red, pooled lines, for individual lines refer to Fig S2). Horizontal grey lines at 5 and −5 indicate significance thresholds. Note the nearly unaltered trafficking in FUS and TDP43 at the proximal readout as opposed to strong negative parameter deviations at the distal site in FUS distinct from the more modest phenotype of TDP43. Conversely, C9ORF exhibited a flat line similar to control lines, consistent with no phenotype at D21 (Fig 2A and C). **(C)** Hierarchical cluster dendrogram of entire signatures shown in (B). The hierarchical Z-score (ordinate) indicates the deviation of entire signatures from each other and is not to be mistaken with the individual parameter Z-scores in (B). Blue boxed cluster highlights physiological signatures. Note how the phenotypically unremarkable C9ORF (red) clustered with Ctrl (blue) against the deviate TDP43 (green) and more severe FUS mutant (grey). **(D)** Hierarchical cluster dendrogram of partial signatures comprising either only all distal or proximal parameters (Mito- and LysoTracker, respectively) to compare site-specific phenotypes. Note how both Ctrl parts (blue; distal and proximal) clustered closely with the proximal FUS part (grey) on the right into a physiological super cluster boxed in blue because of the close physiological trafficking state, whereas the drastic organelle arrest in the distal FUS part on the far left (grey) was highly distinct to its physiological parts at the proximal site. TDP43 showed some moderate deviation in its proximal part (green) within the physiological super cluster and a clear deviation in its distal part (green), albeit less drastic than FUS (grey). C9ORF was contained in the physiological super cluster with both the distal and proximal parts because of no phenotype at D21.

generated a hierarchical cluster dendrogram with KNIME, as described (Pal et al, 2018) (Fig 1C and D). As expected, FUS and TDP43 were each assigned to distinct clusters, with FUS being even more deviated, whereas C9ORF and Ctrl cells were grouped together in a cluster termed "physiological" (Fig 1C and D), consistent with no observed cross-phenotype on D21 (Figs 1B, 2A and C, S1, and S2 and Videos 1 and 2). In summary, hierarchical clustering confirmed the distinct phenotype of mutant FUS and TDP43 confined to the distal

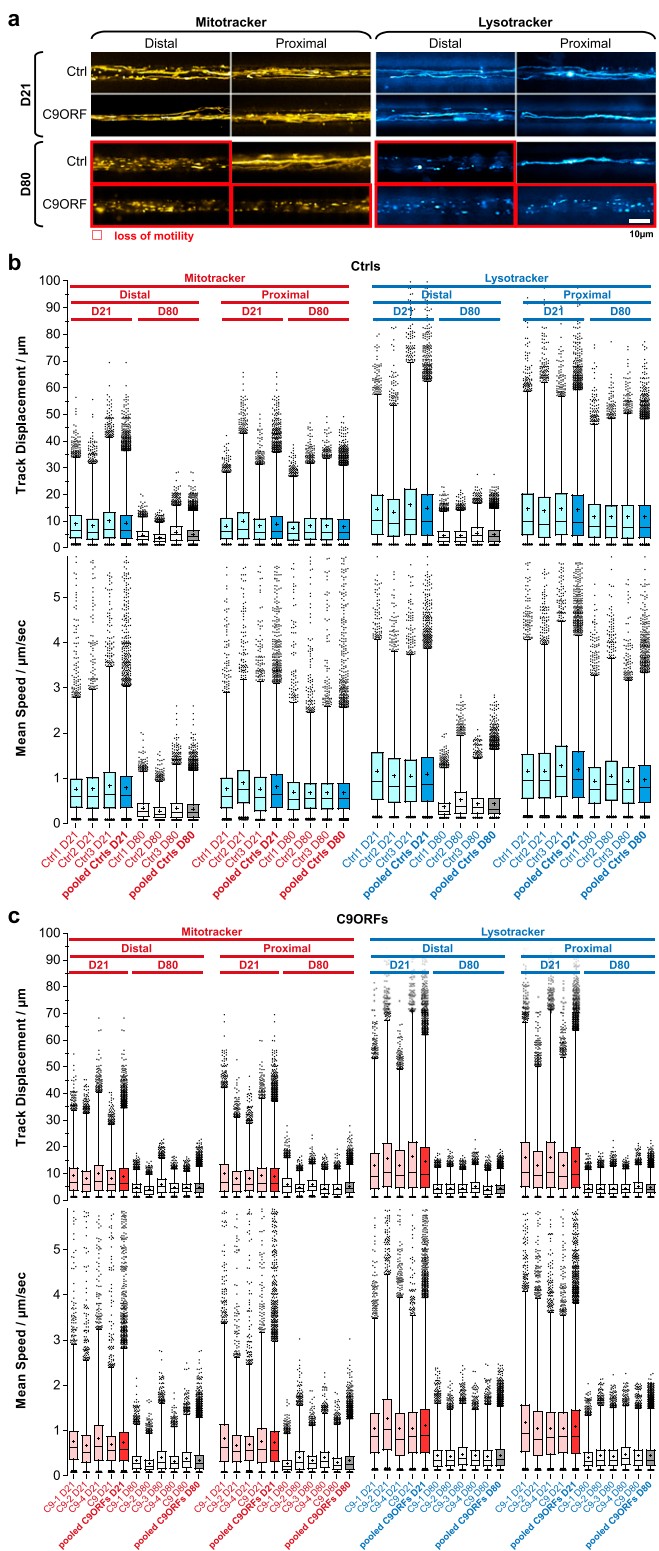

**Figure 2. Loss of organelle motility in aged C9ORF72 spinal MNs.**
**(A)** Maximum intensity projections of movie raw data acquired live with MitoTracker (left) and LysoTracker (right) at the distal (left) versus the proximal (right) microchannel readout position as shown in Fig 1A. Movies were acquired on day 21 during maturation (top galleries, D21) versus aged stage on D80 (bottom galleries). Red-boxed images highlight loss of motility. Note the loss of motility

axon as opposed to the lack or very small alteration in C9ORF on D21.

## Organelle trafficking defects deteriorate during ageing in HRE *C9ORF72*

Different from FUS-ALS, C9ORF72-ALS is a classical old onset form of ALS (Millecamps et al, 2012). Thus, we hypothesized that degenerative phenotypes will be visible at later time points and performed fast dual-color live imaging of mitochondria and lysosomes on D21 and D80. All HRE *C9ORF72* lines exhibited normal, physiological distal and proximal organelle motilities on D21 similar to control cells (Figs 2 and S1 and Videos 1 and 2). After ageing until D80, control cells apparently lost their processive organelle motility at the distal site (Figs 2A and S1 and Videos 1 and 2, boxed in red), whereas their proximal trafficking appeared unaltered feasibly because of a "physiological" retrograde dying back of axons over extended ageing (Naumann et al, 2018). Conversely, all HRE *C9ORF72* lines exhibited a decline in trafficking on D80 at both the distal and proximal sites, suggesting a distinct progression of neuro-degeneration (Figs 2 and S1 and Videos 1 and 2, boxed in red).

For each organelle type, we displayed mean speed and track displacement resulting from the tracking analysis as box plots (Fig 2B and C). All control (Fig 2B) and HRE *C9ORF72* (Fig 2C) lines exhibited, in essence, two rather discrete trafficking states: one mobile (blue boxes for all control lines, Fig 2B; red boxes for all HRE *C9ORF72* lines, Fig 2C) with an average track displacement around 9/14 μm (Mito-/LysoTracker) and mean speed of about 0.7/1.1 μm/s (Mito-/LysoTracker) as opposed to one relatively immobile (grey boxes, Fig 2B and C) with an average track displacement around 5/5 μm (Mito-/LysoTracker) and mean speed of 0.4/0.4 μm/s (Mito-/LysoTracker). On D21, all lines (control and HRE *C9ORF72*) displayed the mobile state at either site (distal and proximal) for either type of organelle (mitochondria and lysosomes). On D80, all control lines had their trafficking still fully maintained in the mobile state at the proximal axon site but deteriorated to the immobile state at the distal site (Fig 2B), whereas all HRE *C9ORF72* lines showed the immobile state at both the distal and proximal sites (Fig 2C), consistent with the apparent pathological decline in the raw data (Figs 2A and S1 and Videos 1 and 2).

on D80 in both distal and proximal C9ORF axons as opposed to distal loss only in Ctrl neurons. Shown are the Ctrl2 and C9-1 lines (Table 2) as representative examples. An overview of all lines is provided in Fig S1. Scale bar = 10 μm. **(B)** Organelle tracking analysis corresponding to (A) of all Ctrl lines as box plots, MitoTracker (left), and LysoTracker (right) distal versus proximal and D21 versus D80 as indicated in the header. Whiskers: 1–99%, box: 50%, horizontal line: median, cross: mean, outliers: black dots. Shown are the individual Ctrl lines (Ctrl1–3, in pale blue or light grey) along with the pooled analysis (Ctrls pooled, in full blue or dark grey), as indicated in bottom labels. The tracking analysis was performed for organelle track displacement (top box plots) and mean speed (bottom box plots). Physiological motility is indicated in pale/full blue as opposed to organelle arrest in light/dark grey. Note that proximal motility remained physiological over ageing (D21 and D80) in all control lines as opposed to the distal decline on D80. **(C)** Same as (B) but for all C9ORF lines. Shown are the individual lines (C9-1, C9-2, C9-3, C9-4, and C9 in pale red or light grey) along with the pooled analysis (C9ORFs pooled, full red or dark grey), as indicated in bottom labels. Physiological motility is indicated in pale/full red as opposed to organelle arrest in light/dark grey. Note that both distal and proximal motilities were lost over ageing (compare D21 and D80) in all C9ORF lines, whereas proximal control organelles in (B) remained motile over ageing. C9-3 was not measured on D21.

**Table 2.  List of abbreviations in alphabetical order.**

| Abbreviation | Alias | Full term |
|---|---|---|
| 53BP1 | | Tumor suppressor 53–binding protein 1 and DSB marker |
| ALS | | Abramyotrophic lateral sclerosis |
| ASO | | Antisense oligonucleotide |
| ATG | | Adenine–thymine–guanine start codon of translation |
| BNS-22 | | DNA topoisomerase 2 inhibition without causing SSBs or DSBs |
| C9 | C9ORF, C9ORF72 | Chromosome 9 open reading frame 72 |
| C9-GC | | Gene-corrected C9 isogenic cell line, that is, with HREs in intron 1 of *C9ORF72* gene excised |
| C9-KO | | Isogenic C9 cell line with functional KO of the exonic, ATG-transcribed *C9ORF72* gene part with HREs in intron 1 preserved |
| Campto | | Camptothecin, DNA topoisomerase 1 inhibitor, and SSB inducer |
| Cas9n | | CRISPR-associated 9 nickase |
| Casp3 | | Apoptosis marker cleaved caspase 3 |
| CRISPR | | Clustered regularly interspaced short palindromic repeats |
| Ctrl | | Wild-type control cell line |
| D | e.g. D14, D21, D80 | Days after seeding of MNs into MFCs |
| DEG | | Differentially expressed gene |
| DMSO | | Dimethylsulfoxide, common solvent for chemcial compounds, inhibitors, etc. |
| DPR | | Dipeptide repeat protein, encoded by RAN translation of intronic HREs |
| DSB | | DNA double-strand break |
| ELISA | | Enzyme-linked immunosorbent assay |
| Eto | | Etoposide, DNA topoisomerase 2 inhibitor, and DSB inducer |
| FISH | | Fluorescence in situ hybridization |
| FUS | | Fused in sarcoma |
| GGGGCC | | Guanine(x4)-cytosine(x2) hexanucleotide sequence motif of HREs |
| GOF | | Gain of (toxic) function |
| HC | | High content |
| hnRNP | | Heterogeneous ribonucleoprotein |
| HRE | | Hexanucleotide repeat expansion in intron 1 of *C9ORF72* gene |
| HSP | | Heat shock protein |
| iPSC | | Induced pluripotent stem cell |
| KD | | Gene knockdown |
| KO | | Gene knockout |
| LNA | | Locked nucleic acid nucleotides, used in RNA FISH probes |
| LOF | | Loss of function |
| LUT | | Look up table |
| MAP2 | | Microtubule-associated protein 2 and dendritic neuron marker |

**Table 2. Continued**

| Abbreviation | Alias | Full term |
|---|---|---|
| MFC | | Xona® microfluidic chamber, for compartmentalized neuron cultures with axons aligned in microchannels with defined orientation with respect to somata and clear physical separation from dendrites |
| MN | | Motoneuron |
| nHR | | Normal hexanucleotide repeats |
| NPC | | Neuronal progenitor cell |
| OXPHOS | | Oxidative phosphorylation, referring to the mitochondrial respiratory ATP production |
| pATM | | Phosphorylated ataxia telangiectasia mutated protein |
| poly GA | GA | Poly glycine–alanine DPR, one species in the spectrum of RAN-translated HREs |
| poly GP | GP | Poly glycine–proline DPR, one species in the spectrum of RAN-translated HREs |
| poly PR | PR | Poly proline–arginine DPR, one species in the spectrum of RAN-translated HREs |
| PPI | | Protein–protein interaction |
| RAN translation | | Repeat-associated non-ATG translation |
| R-loops | | Three-stranded hybrid structure composed of DNA template and complementary strand and nascent mRNA during transcription |
| ROS | | Reactive oxygen species |
| RPS25 | | Ribosomal protein subunit of 25 kDa |
| RRE | | RNA repeat expansion |
| SSB | | DNA single-strand break |
| TDP43 | | TAR DNA-binding protein-43 |
| WT | | Wild type |
| γH2AX | | Phosphorylated histone H2A.X, DSB marker |

## Multiparametric spatiotemporal HC organelle tracking analysis revealed a proximal axonopathy in HRE *C9ORF72*

In light of the distal axonopathy in mutant FUS and TDP43 already on D21 along with premature axonal dying back (Naumann et al, 2018) and a similar figure in physiological control cells on D80 (Figs 2A and B, S1, and S2 and Videos 1 and 2), we envisioned two distinct plausible scenarios to explain the virtual organelle arrest at both the distal and proximal sites in C9ORF72 over extended ageing on D80 (Figs 2A and C, S1, and S2 and Videos 1 and 2): either dying back of axons progresses earlier and faster between D21 and D80, thereby reaching the proximal readout until D80, whereas in control cells, the slower dying back has still no impact here, or the proximal phenotype occurs independent of axonal dying back by a distinct mechanism.

To understand these spatiotemporal differences in more detail, we analyzed the multiparametric HC signatures at multiple time points over a course from D14 to 80 (Figs 3A and S2). Given the high similarities within Ctrl and C9ORF lines, respectively, Fig 3 shows pooled signatures (Ctrl: dark blue; C9ORF: red), whereas Fig S2 provides the corresponding profiles of individual lines. On D14, 21, and 28, Ctrl and C9ORF lines displayed virtually flat lines as signatures, consistent with no cross-phenotype until D28 (Fig 3A). However, from D40 onward, we observed negative deviations in distal control axons (dark blue profiles, Fig 3A) for most speed parameters, track duration, and displacement for either type of organelle. The overall reduction in organelle motility further exacerbated but remained restricted to the distal site in controls until D80 (dark blue profiles, Fig 3A). Only few proximal lysosome parameters finally showed borderline significance in their alterations that were, however, marginal compared with the drastic impairments at the distal site.

By contrast, C9ORF cells exhibited such trafficking defects always simultaneously at both the distal and proximal sites with an onset on D40 (red profiles, Fig 3A). The further progression of these defects appeared unsteady in some signature parts. Particularly for proximal lysosomes, we observed a rapid parameter decrease from D40 to 50, followed by some transient stagnation on D60 toward most severe impairment on D80 (red profiles, Fig 3A). The concurrent emergence of trafficking defects at both the distal and proximal sites from D40 onward was a surprising finding arguing against a simple retrograde dying back of the axon but pointing toward a proximal axonopathy by a distinct mechanism.

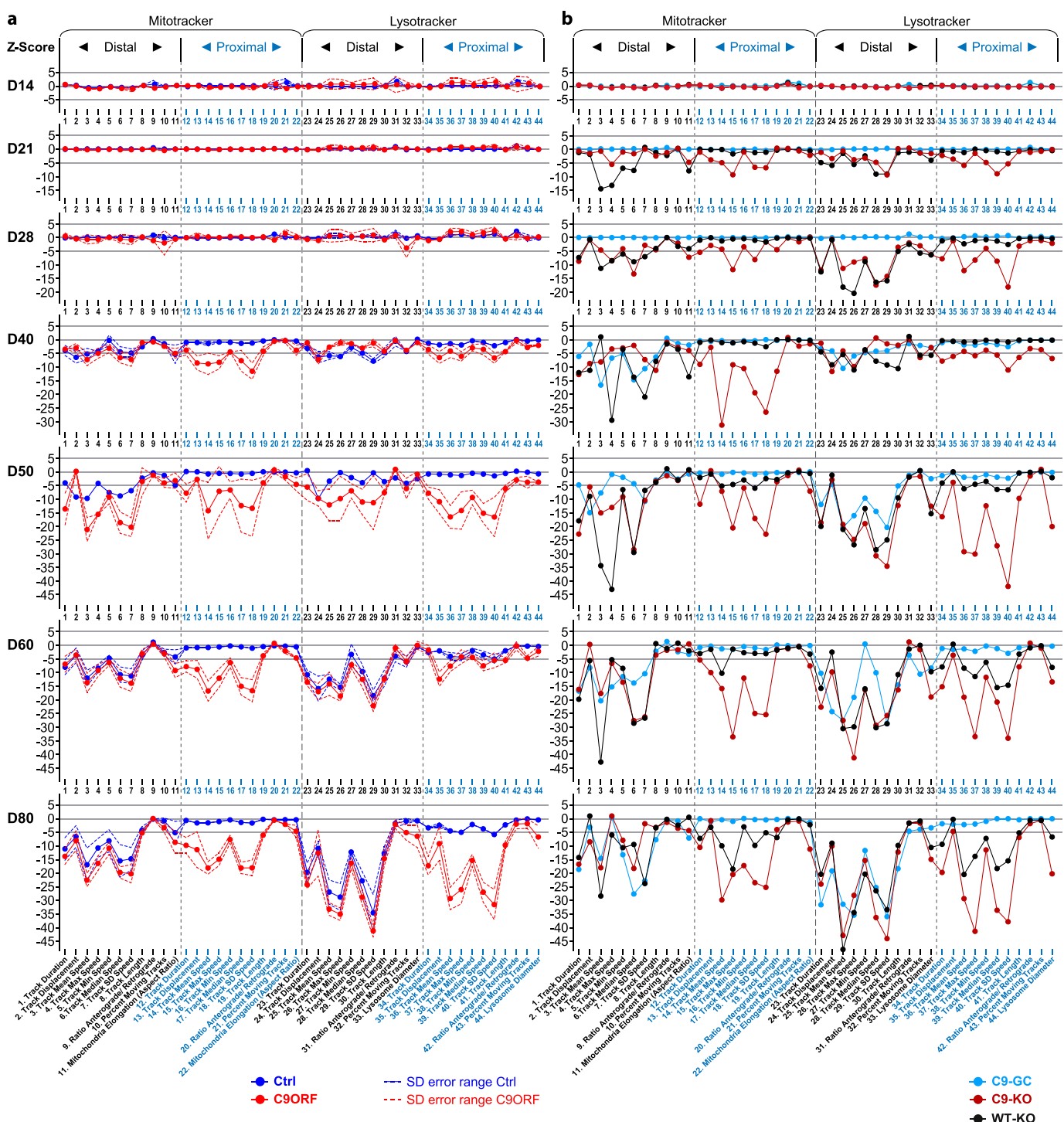

**Figure 3. High-content phenotypic profiling over extended time course revealed global axonal trafficking defects in C9ORF72 spinal MNs over ageing.**
**(A)** Multiparametric high-content profiles of pooled Ctrl (in blue) and C9ORF lines (in red) were deduced as in Fig 1B over a time course from D14 to D80, as indicated on the left. (For overview of all individual lines refer to Fig S2.) Z-scores for all time points were calculated with respect to pooled control lines at D21 proximal. Dotted lines indicate error ranges (SD between lines). Note the onset of trafficking decline (negative parameter deviations, and Z-scores ≤ −5) for either organelle type (Mito- and LysoTracker) from D40 onwards that progressed only at the distal readout site in control lines as opposed to the global phenotype in C9ORF72, that is, simultaneous emergence at both the distal and proximal sites. **(B)** Similar time course for (i) C9-GC with excised intronic hexanucleotide repeat expansions (HREs) (in light blue), (ii) C9-KO with intronic HREs preserved but with KO of the exonic C9ORF72 part (in brown), and (iii) WT-KO cells with the same exonic KO and naturally having no intronic HREs (in black). Note how the proximal trafficking decline in C9-GC was restored to physiological levels over the entire time course, whereas the distal decline remained unaltered (compare light blue profiles with dark blue counterparts in (A)). As for C9-KO, note the earlier onset of global (distal and proximal) trafficking defects already on D21 (Z-scores ≤ −5) compared with D40 in parental C9 (compare brown profiles with red counterparts in (A)). As for WT-KO, note the earlier onset of distal trafficking defects already on D21 and the emergence of a proximal decline at later time points as well (from D50 onwards) as opposed to distal decline only in parental Ctrl (compare black profiles with dark blue counterparts in (A)).

## Both gain and loss of functions contribute to trafficking deficiency in C9ORF72

There is extensive discussion about the pathomechanisms of *C9ORF72* HRE including mainly gain of toxic functions (GOFs) (Walker et al, 2017; Nihei et al, 2020) versus loss of C9ORF72 functions (LOFs) (Farg et al, 2014; Shi et al, 2018). Thus, we wished to investigate the role of GOF and LOF mechanisms in HRE-mediated trafficking defects. To this end, we used CRISPR/Cas9n–mediated gene editing on one parental HRE line, namely, C9 (Table 2), to generate an isogenic set of additional MN models. A full description and characterization of these KO and GC variants was recently published (Abo-Rady et al, 2020) (Table 2), including a verification of eliminated C9ORF72 expression in all KO variants and of DPRs in the GC variant. We deduced multiparametric HC signatures for the isogenic C9-GC, C9-KO, and WT-KO lines (Table 2) over the same time course from D14 to 80 (Fig 3B).

We first analyzed a gene-corrected control line (C9-GC) with excision of the intronic HREs by replacing the HRE in intron 1 with its wild-type counterpart of only 1–3 repeats. This led to a full restoration of all proximal parameter deviations for either type of organelle at any time, thereby resulting in wild-type–like signatures with a similar onset on D40 and further progression of distal trafficking defects until D80 without proximal phenotypes (compare light blue profiles in Fig 3B with dark blue in Fig 3A). Hence, gene correction indicated that all phenotypic perturbations over extended ageing in our C9ORF lines were truly due to the *C9ORF72* HRE mutation.

Because it is known that HRE-expressing C9ORF72 causes reduced expression of C9ORF72 (Waite et al, 2014; Sivadasan et al, 2016; Frick et al, 2018), we cannot distinguish from these results whether GOF or LOF caused the trafficking phenotypes. To address this, we abolished C9ORF72 expression by excision of the translation start codon in exon 2 as effectively as a conventional gene KO, except that HRE-mediated DPR expression from intron 1 still occurred to study their GOF in the absence of C9ORF72 (C9-KO). KO of exonic *C9ORF72* while maintaining HREs (C9-KO) led to a premature onset of negative parameter deviations already on D21 (compare brown profiles in Fig 3B with red in Fig 3A), thereby exacerbating the natural C9ORF phenotype.

Having established that KO of exonic *C9ORF72* is exaggerating C9ORF72 HRE phenotypes, we asked whether KO of exonic *C9ORF72* in control cells (WT-KO) without HREs is sufficient to cause any phenotypes. By using the same KO method on a healthy control line (Ctrl1, Table 2), we abolished C9ORF72 expression in a wild type with naturally no HREs to study C9ORF72 LOF in the absence of RAN-translated DPRs (WT-KO). This led to a premature onset on D21 as well, but initially only at the distal site (compare black profiles in Fig 3B with dark blue in Fig 3A). Remarkably, we observed a delayed onset of proximal impairments as well from D50 onward, finally resulting in a signature on D80 resembling the natural C9ORF profile (compare black profiles in Fig 3B with red in Fig 3A). This peculiar spatiotemporal appearance of this axonal phenotype might suggest that the dying back of axons progresses earlier and faster between D21 and D80, thereby reaching the proximal readout until D80, whereas in control cells, the slower dying back has still no impact here. In summary, these results indicate that both the GOF of HRE/DPRs and LOF of C9ORF72 protein mechanisms contribute through combinatorial action to the overall C9ORF phenotype.

Because altered onsets and site-specificities (i.e., distal versus proximal axon sites) were emerging as major phenotypic perturbations, we sought to extract these distinctive features from our multiparametric data sets at better clarity. To this end, we summed up the absolute values of all distal versus proximal parameter Z-scores (for both the Mito- and LysoTracker, respectively) to obtain a site-specific measure of overall phenotypic strength relative to control baseline along with the total phenotypic strength of the entire signature. The resultant phenotypic strength values were plotted over time that yielded no error bars (Figs S3 and S4) because the underlying Z-scores were already based on pooled data sets across all experiments and calculated on the base of standard deviations, thereby not allowing for "new" or "extra" error bars (exception: graphs for pooled lines). Therefore, the significance in the increase in phenotypic strength over time was already intrinsic in the increase per se. Clearly, Ctrl cells did hardly show any increase in their proximal phenotypic strength over ageing and remained on the initial base level (in blue, top gallery, Fig S3), consistent with no visible phenotype (Figs 2A and S1 and Videos 1 and 2) and no appreciable proximal parameter deviations in the corresponding signatures (Figs 3A and S2). By contrast, the distal phenotypic strength was steadily increasing from a flat line in Ctrl cells from D40 onwards (in red, top gallery, Fig S3), thereby clearly marking the onset of distal trafficking impairments. C9ORF cells exhibited an increase at both the distal and proximal sites (compare blue with red, bottom gallery, Fig S3) from D40 onwards (red profiles, Fig 3A). Excision of intronic HREs (C9-GC) resulted in a flat line at the proximal site over the entire time course indistinguishable from Ctrl cells, consistent with full restoration because of the gene correction (in blue, bottom gallery, Fig S3). Conversely, KO of exonic *C9ORF72* with intronic HREs preserved (C9-KO) led to an earlier increase in phenotypic strength from D21 onward simultaneously at both axon sites (compare blue with red, bottom gallery, Fig S3). Finally, KO of exonic *C9ORF72* in control cells with naturally no HREs (WT-KO) resulted in a premature increase at the distal site, whereas the proximal increase was delayed until D50 (compare blue with red, top gallery, Fig S3). In summary, the plotting of site-specific phenotypic strength over ageing confirmed the differences in the spatiotemporal progression of axonal trafficking defects at high clarity.

## Axonal trafficking defects in ageing C9ORF72 MNs were due to common, systemic perturbations of the cytoskeleton and energy supply

To gain further mechanistic insights, we were finally addressing the underlying cause for the compromised axonal motility of mitochondria versus lysosomes bioinformatically (Fig 4). Our C9ORF lines were recently generated and characterized (Abo-Rady et al, 2020). The authors performed transcriptomics by deep sequencing, as highlighted in Fig 3 in that publication (Abo-Rady et al, 2020). The obtained gene ontology terms pointed to many alterations of microtubules and motors, DNA damage response, and apoptosis, but not to anything specific for mitochondria and/or lysosomes. This provided a hint that the trafficking defects as seen by us are likely due to a more

upstream, presumably systemic, cause that impacts accordingly on different organelle types in a similar fashion. To further refine the previous transcriptomics (Abo-Rady et al, 2020), we retrieved the original RNA-seq data set GSE143743 to recalculate lists of differentially expressed genes (DEGs) for C9 versus C9-GC and C9 versus C9-KO (Fig 4A, see the Materials and Methods section for details). Next, we built interactome maps (Fig 4B–F), based on known protein–protein interactions (PPIs) encoded by the DEGs using STRING (version 11.0, https://string-db.org/) (Szklarczyk et al, 2019). These interactome maps were further divided into subnetworks of proteins, each of which represented potential functional clusters or functional modules (using MCL clustering, encircled in Fig 4B–F). To account for the possibility that not all interactors (i.e., proteins and nodes) were showing up as DEGs, we performed further refinement of the primary interactome maps (Fig 4B and D) by enabling up to 50 additional direct known interactors to the original DEG-encoded nodes (i.e., the "first shell") and up to 50 additional interactors to the "first shell" (i.e., indirect "second shell" interactors), that is, a total of 100 additional known interactors to fill missing nodes for better identification and visibility of functional clusters (Fig 4C, E, and F, refer to the Materials and Methods section for more details). The comparison of C9 versus C9-GC revealed 2,586 DEGs (both up- and down-regulated), whereas C9 versus KO revealed 581 DEGs (Fig 4A). Comparing C9 versus C9-GC revealed all disease-specific HRE-mediated (GOF) DEGs, whereas C9 versus C9-KO revealed only LOF-specific DEGs. Both pairwise comparisons showed an overlap of 427 DEGs, referring to HRE-mediated LOFs (Fig 4A), that is, these common DEGs pointed to an HRE-mediated upstream mechanism that caused in turn a LOF of exonic C9ORF72. Remarkably, of the 581 LOF-specific DEGs (C9 versus C9-KO) 73% (427) were shared with the GOF-specific DEGs of C9 versus C9-GC (Fig 4A), thereby suggesting that the great majority of LOF-specific DEGs were due to an upstream HRE-mediated GOF. The 2,586 DEGs of C9 versus C9-GC revealed three functional clusters in the PPI interactome map (Fig 4B), annotated with (i) ribosome, (ii) focal adhesion, AMPK signalling, and (iii) microtubule network, cytoskeletal filaments, Rho GTPase. Further refinement by limiting to down-regulated DEGs only and enabling 100 additional known interactors (Fig 4C) revealed four functional clusters annotated with (i) RNA polymerase, (ii) endocytosis, neuroactive ligand–receptor interaction, (iii) mitochondrial oxidative phosphorylation (OXPHOS), and again (iv) microtubule-associated proteins, chromatin regulation. The 581 DEGs of C9 versus C9-KO revealed three functional clusters in the PPI interactome map (Fig 4D), annotated with (i) nervous system, endocytosis, (ii) ECM–receptor interaction, MAPK signalling, and again (iii) microtubule-associated proteins, actin cytoskeletal network. Further refinement by limiting to down-regulated DEGs only and enabling 100 additional known interactors (Fig 4E) revealed four functional clusters annotated with (i) apoptosis, (ii) neuronal vesicular transport/SNARE, (iii) MAPK signalling, transcription, and again (iv) mitochondrial OXPHOS. Finally, the 427 DEGs common to both GOF and LOF with 100 additional known interactors enabled (Fig 4F) revealed three functional clusters annotated with (i) focal adhesion, ECM–receptor interaction, again (ii) microtubule-associated proteins, DNA repair, and again (iii) mitochondrial OXPHOS. The common denominators from these pairwise comparisons and their refinements clearly pointed to alterations in microtubule-associated interactions and mitochondrial OXPHOS, thereby further supporting a systemic cause of axonal trafficking defects in C9ORF72 through impaired cytoskeleton-associated interactions and energy deprivation further upstream that affected both the motility of mitochondria, lysosomes, and presumably other organelle types. Noteworthy is the annotation of endocytosis (Fig 4C and D). This might point to the known role of C9ORF72 in endocytosis with its RNAi-mediated KD impacting on lysosomal degradation and autophagy (Farg et al, 2014).

## HRE-mediated axonal organelle trafficking defects concurred with DPR and DNA damage accumulation along with apoptosis

Two commonly recognized hallmarks in the pathology of neurodegeneration are nuclear DNA damage accumulation and apoptosis (Madabhushi et al, 2014; Naumann et al, 2018). Specifically in C9ORF ALS, HRE-based pathology is believed to be mediated through RAN-translated DPRs (Chew et al, 2015; O'Rourke et al, 2015; Jiang et al, 2016). Thus, we sought to score directly for them with respect to DNA damage and apoptosis when axonal trafficking defects emerged. To this end, we performed immunofluorescence confocal microscopy on the isogenic lines at the end of our time course on D80 versus D21 (i.e., before and after emergence of trafficking defects, Fig 3) to reveal the prominent DPR variants poly GP and poly GA (Nihei et al, 2020) along with the DSB markers phospho-histone H2A.X (γH2AX, Figs 5 and S6) and tumor suppressor 53–binding protein 1 (53BP1, Figs 6, S5, and S6) as well as the apoptosis marker cleaved caspase 3 (Casp3, Fig 7A), as described (Naumann et al, 2018). As for other DPR variants (e.g., GR), we failed to obtained specific staining patterns with available antibodies and rejected them from this study (data not shown). Control cells did only show traces of GP, GA, and either DSB marker at both time points (Ctrl1, Figs 5 and 6). By contrast, parental C9ORF cells and their KO counterpart exhibited a drastic accumulation of both DPR variants on D80 (C9 and C9-KO, Figs 5 and 6), whereas on D21, they were indistinguishable from control cells (Ctrl1, Figs 5C, 6B, S5, and S6). Specifically for GP, the accumulation occurred as aligned neuritic foci in MAP2-positive neurons (Figs 5A and S6, green arrowheads). Conversely for GA, the accumulation occurred as larger perinuclear foci (Fig 6A, green arrowheads), consistent with histological brain sections in C9ORF72 patients in a recent report (Nihei et al, 2020). Accumulation of both DPR variants concurred with augmented DSBs, (Figs 5A and 6A, white arrowheads). In case of GP, DSBs were revealed with γH2AX and in case of GA, with 53BP1 antibodies because of a species conflict in the co-staining cocktail (see the Materials and Methods section). However, we verified that both DSB markers revealed very similar staining patterns in all lines (Fig S7A, yellow arrowheads) with high colocalization when DSBs augmented (Fig S7B and C).

Remarkably, KO of exonic C9ORF in control cells with naturally no HREs mimicked DSB accumulation as high as in C9 or C9-KO lines (WT-KO, Figs 5A and B, 6A, and S7A–C), suggesting that LOF of C9ORF72 is the driving factor for appearance of DSBs, rather than augmented DPRs. By contrast, gene correction of HREs in C9ORF reverted augmented GP, GA, and DSBs back to control levels (C9-GC, Figs 5, 6, and S7), consistent with the axonal trafficking profiles (Fig 3B).

The concurrence of augmented DPR and DSB foci in C9 and C9-KO on D80 (Figs 5 and 6) raised the question whether either type of perturbation concurred in the same neuron or in two distinct parts of the population. This presents an important aspect of C9ORF72

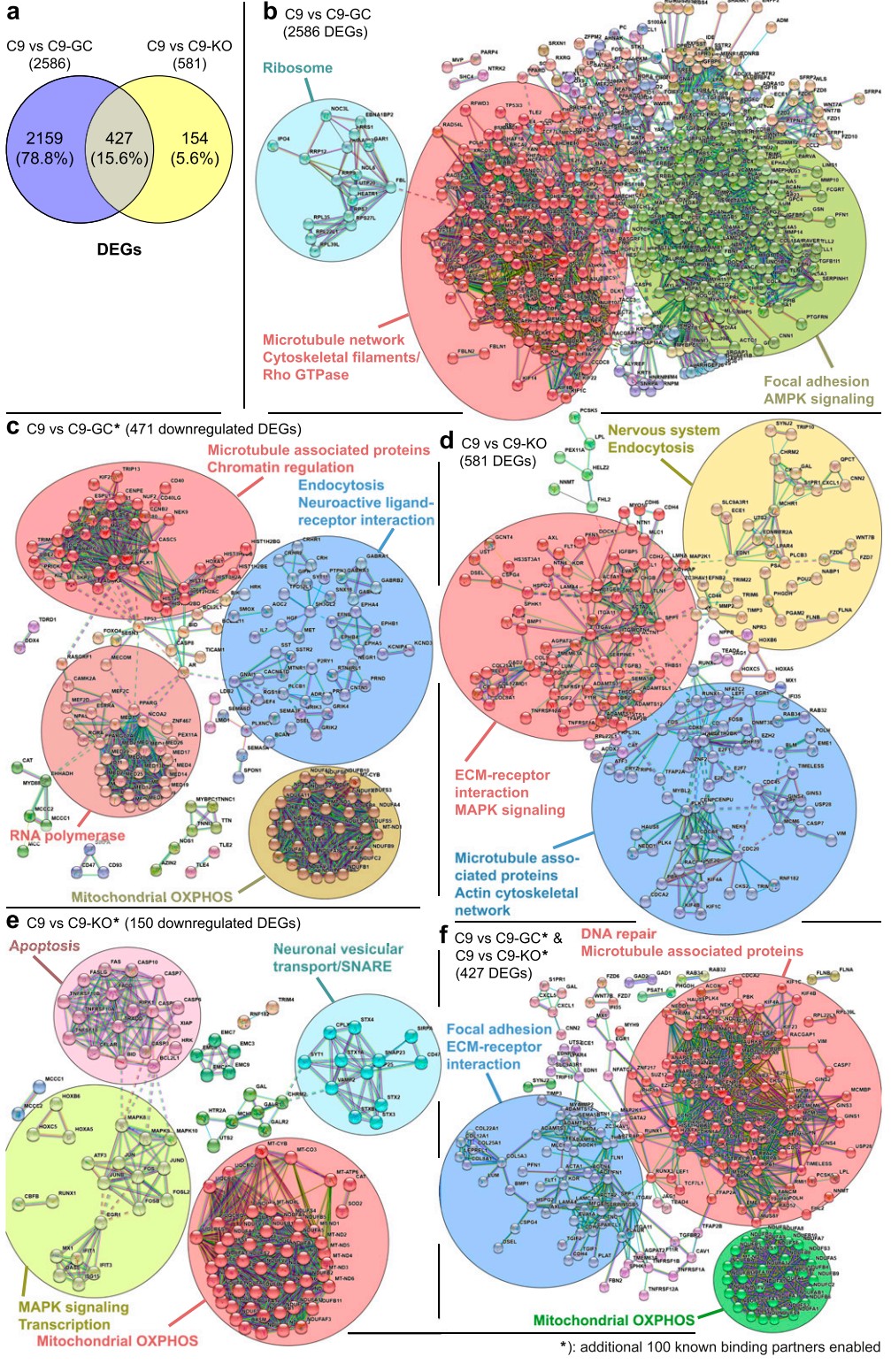

**Figure 4. Identification and functional clustering of differentially expressed genes (DEGs) from high-throughput RNA-seq data sets.**
**(A)** Venn diagrams showing the number of DEGs between C9 versus C9-GC spinal MNs (purple), between C9 versus C9-KO (yellow), and the observed overlap of both comparisons (kahki). **(B, C, D, E, F)** Protein–protein interaction (PPI) network analysis of the DEGs in C9 versus C9-GC (B, C), C9 versus C9-KO (D, E) and the common DEGs shared by both comparisons (F), as illustrated in (A). The nodes (bubbles) indicate the DEGs and the connecting lines the interaction between two proteins. The STRING database (Szklarczyk et al, 2019) was used to establish the interaction network, with a highest confidence score of >0.9 (STRING scores > 0.900). MCL clustering (inflation = 1.5) was applied on the PPI network to select most significant functional clusters or subnetworks. Clusters of functionally related nodes were manually encircled and annotated. Disconnected nodes were omitted. Statistical significance of P-value < 0.05 was applied in the network. (Refer to Tables S1–S5 for complete list of nodes.) **(B)** PPI network analysis of 2,586 DEGs C9 versus C9-GC, as illustrated in (A) revealed three functional clusters of hexanucleotide repeat expansion (HRE)–mediated, disease-specific interacting partners. Given the high number of seed proteins, a zero-order interaction network was performed using the NetworkAnalyst tool (Xia et al, 2015). No additional known interactors were enabled. **(C)** Further refinement of (B) restricted to 471 down-regulated DEGs and with 100 additional known interactors enabled (i.e., 50 direct "first shell" and 50 indirect "second shell" interactors, see the Materials and Methods section for details) revealed four functional clusters. **(D)** PPI network analysis of 581 DEGs C9 versus C9-KO as illustrated in (A) without additional known interactors revealed three functional clusters of C9ORF72 LOF-mediated interacting partners. **(E)** Further refinement of (D) restricted to 150 down-regulated DEGs and with 100 additional known interactors enabled revealed four functional clusters. **(F)** PPI network analysis of 427 DEGs shared by both comparisons (C9 versus C9-GC and C9 versus C9-KO), as illustrated in (A) with 100 additional known interactors enabled revealed three common functional clusters. **(B, C, D, E, F)** Summary and conclusion: note the identification of cytoskeletal- (mostly microtubule) related functional clusters in both HRE- (B, C) and LOF-mediated

DEGs (D) of which many were shared by both comparisons (F), thereby pointing to a common, systemic microtubule-based underlying cause of axonal trafficking defects (Fig 3) that affects several organelle types in the same way. (C, E, F) Likewise, further refined PPI network analysis (C, E) revealed functional clusters for mitochondrial oxidative phosphorylation (OXPHOS) for both HRE- (C) and LOF-mediated (E) DEGs of which many were shared by both comparisons (F), thereby pointing to a common, systemic energy deprivation as further underlying cause of axonal trafficking defects (Fig 3).

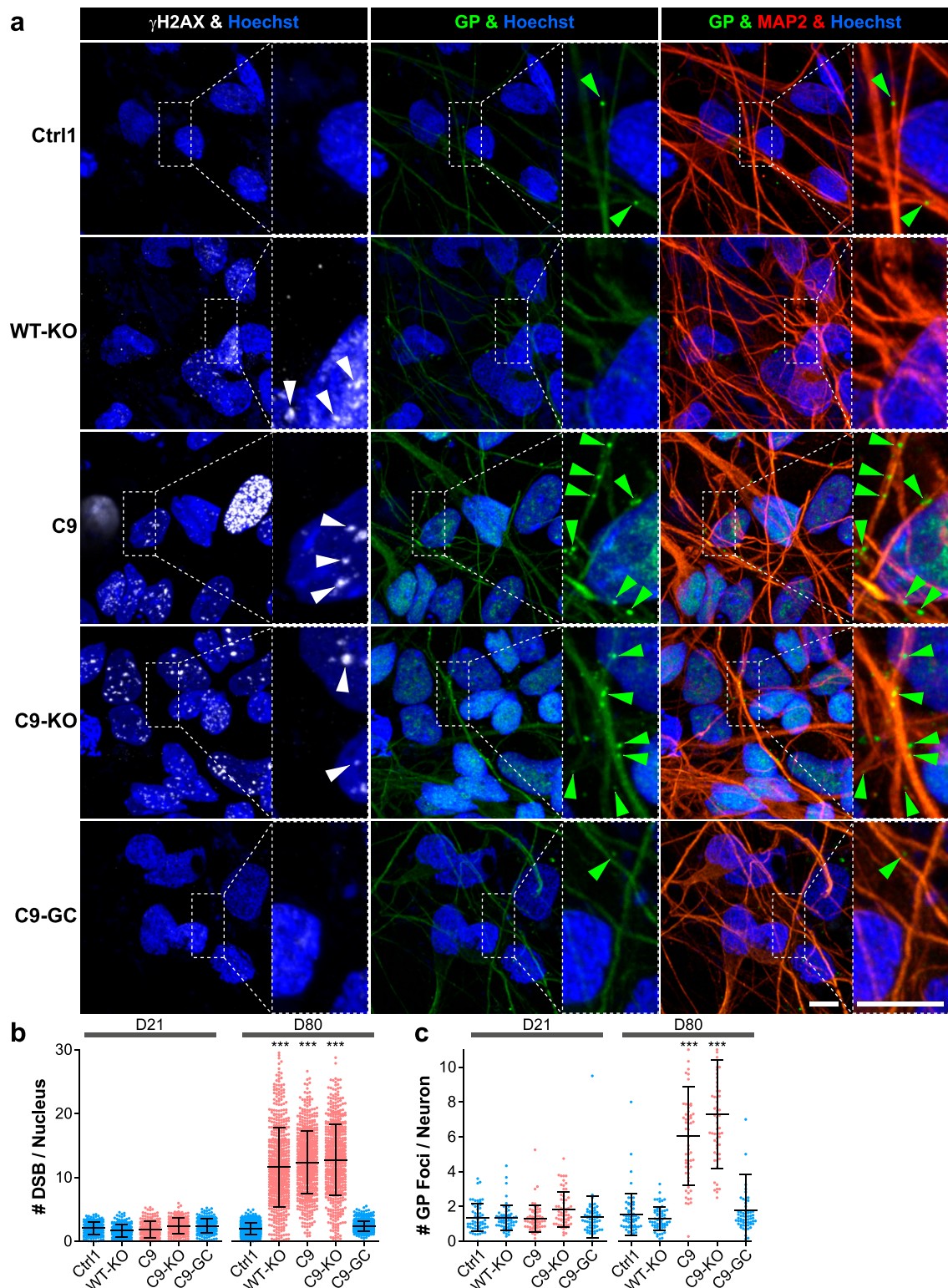

**Figure 5.    DNA damage accumulation concurred with neuritic glycine–proline (GP) dipeptide repeat protein foci in C9ORF72 spinal MNs over ageing.**
**(A)** DNA double-strand break (DSB) marker γH2AX (in white) in Hoechst-positive nuclei (in blue) and neuritic GP foci (in green) in MAP2-positive neurons (in red) were revealed by confocal IF microscopy at D80 endpoints (Fig 3). Dotted boxed areas in image galleries are shown magnified on the right. Note the striking nuclear accumulation of γH2AX-positive nuclear foci (white arrowheads) in parental C9 and C9-KO that were phenocopied by WT-KO. Furthermore, GP foci aligned to neurites (green arrowheads) concurred with nuclear γH2AX accumulation in C9 and C9-KO. Conversely, DSBs and GP foci were nearly absent in C9-GC and parental Ctrl1. Arrowheads point only to arbitrary examples. Scale bars = 10 $\mu$m. **(B)** Quantification of DSBs in (A) in MAP2-positive neurons (count of γH2AX-positive foci per nucleus) on

pathology as the causative GOF of elevated DPR expression in DNA damage is hotly debated (Walker et al, 2017; Nihei et al, 2020). As for the neuritic GP foci, we were unable to assign affected neurites to their respective somata in our dense, aged cultures (Fig 5), whereas we always observed augmented nuclear DSBs in the same neuron whenever perinuclear GA foci emerged (Fig 6A and C, green and white arrowheads in A). In essence, ~40% of all C9 and C9-KO neurons displayed perinuclear GA foci together with nuclear DSBs (Fig 6C). In summary, we conclude that DPR accumulation in aged C9ORF spinal MNs is probably always linked to concurrence of DSBs. Besides, DPR accumulation does not seem to be a requirement for DSBs as WT-KO MNs showed DSBs, despite having no HREs and no DPRs.

Finally, axonal trafficking phenotypes and accumulated DSBs in C9, C9-KO, and WT-KO on D80 were mirrored by elevated Casp3 levels, whereas Ctrl1 and C9-GC were showing hardly any sign of apoptosis (Fig 7A). Of note, WT-KO MNs exhibited milder Casp3 levels than parental C9, whereas C9-KO displayed even higher levels (Fig 7B), consistent with the delayed onset of C9-related proximal trafficking defects in WT-KO, as opposed to the premature onset in C9-KO (Fig S3, blue curves).

### TDP43 localization remained unaltered in aged C9ORF72 MNs

Another hallmark in HRE-mediated ALS is the nuclear displacement of TDP43 along with its aggregation in cytosolic inclusion bodies (Neumann et al, 2006; DeJesus-Hernandez et al, 2011; Lee et al, 2011; Scotter et al, 2015), leading to erratic transcription, splicing, sequestration of RNA and RNA-binding proteins, and finally deficient RNA granule transport in axons (Jovicic & Gitler, 2014). Hence, we wished to investigate whether the axonal trafficking defects along with DSB and DPR accumulation on D80 in our C9 lines (Figs 3–7) were linked to TDP43 accumulation. To this end, we revealed TDP43 localization on D80 by IF immunostainings along with γH2AX (Fig S8A). We found, again, an increase in DSBs in WT-KO, C9, and C9-KO with no alteration in the prominent nuclear localization of TDP43 as compared with Ctrl and C9-GC MNs (Fig S8A). Specifically, we determined a marginal pool of nuclei devoid of TDP43 that was indistinguishable across all C9 and Ctrl MNs (Fig S8B). Moreover, a minor pool of TDP43 in cytosolic foci was present in all lines but with no change in foci count (Fig S8C) and total TDP43 amount (i.e., total integral intensity, Fig S8D) across all C9 and Ctrl MNs. In conclusion, the phenotypic perturbations in our C9 cell models were not due to upstream TDP43 mislocalizations.

### DNA damage is not an upstream trigger for axonal trafficking defects in aged C9ORF72 MNs

The concurrence of augmented DSBs and DPRs at D80 endpoints (Figs 5 and 6) raised the question whether these accumulations were actually causative for axonal trafficking defects emerging from D40 onwards (Fig 3). In light of our previous findings in FUS, such mechanistic link appeared plausible. Specifically, through chemical DNA damage induction, we revealed a feasible link between impaired DNA damage response and deficient distal axonal organelle trafficking via a postulated nucleo-axonal cross talk in mutant FUS (Naumann et al, 2018). Likewise, we used different inducers of DNA damage in healthy control C9-GC MNs and performed HC imaging profiling (Fig S9) with the overall question whether DNA damage induction is sufficient to induce a C9ORF72-like HC profile. We used (i) etoposide as DSB inducer through irreversible inhibition of DNA topoisomerase 2 (Pommier et al, 2010), (ii) camptothecin as DNA single-strand break (SSB) inducer through inhibition of DNA topoisomerase 1 (Pommier et al, 2010; Kawatani et al, 2011), and (iii) BNS-22 as negative control as it reversibly inhibits DNA topoisomerase 2 without breaking the DNA (Kawatani et al, 2011). Each inhibitor was added to the proximal MFC site on D21 and 72 h before imaging to enable the required incubation time for the postulated nucleo-axonal cross talk (Naumann et al, 2018). At this time point (D21), FUS did already exhibit its profile (Fig 1B), whereas C9ORF72 did not (Figs 1B, 3, and S2). As a result, we obtained a profile for etoposide reminiscent of FUS (Fig S9A, compare black versus grey profile), that is, with distal deviations only for either type of organelle (Mito- and LysoTracker), consistent with our previous report that did not employ the whole multiparametric profile by that time (Naumann et al, 2018). We confirmed by IF staining that DSB levels were strongly augmented through etoposide treatment (Fig S9B and C). Conversely, camptothecin treatment had only little impact on the control baseline (Fig S9A, compare orange versus light blue profile), suggesting that SSBs had hardly any impact on axonal trafficking. Some mild deviations occurred that were distantly resembling the FUS profile (Fig S9A, compare orange versus grey profile) feasibly because of much fewer DSBs that might occur as secondary DNA damage after the primary SSBs. However, we were unable to detect such secondary DSB augmentation through campto-thecin treatment by IF stainings (Fig S9B and C). Finally, BNS-22 did hardly alter the control baseline (Fig S9A, compare brown versus light blue profile) and did not cause any DSB augmentation (Fig S9B and C), as expected. In essence, none of the DNA damage inducers phenocopied the profile of aged C9 MNs with its characteristic proximal signature parts (Fig S9A, red profile). We also attempted long-term treatments with these inhibitors to test if C9-like profiles would emerge at later time points but encountered massive cell death due to prolonged toxicity. In conclusion, DSBs are unlikely to serve as upstream trigger for axonal trafficking defects in C9 and C9-KO, consistent with a correlative time course analysis of DSBs showing that the onset of DSB augmentation occurred from D60 onward, that is,

---

D21 versus D80 displayed as scatterplots of individual nuclear foci counts with mean (center line) and SD range (whiskers) indicated in black. (For images at D21 endpoint refer to Fig S6.) Note nearly absent DSBs on D21 in all lines versus drastic DSB accumulation on D80 in parental C9, C9-KO, and WT-KO. **(C)** Quantification of (A), number of neuritic GP foci in MAP2-positive neurons on D21 versus D80 displayed as scatterplots of foci counts per neuron and image. (For images at D21, refer to Fig S6.) Note nearly absent GP foci in all lines at D21 versus aligned foci at D80 endpoint in parental C9 and C9-KO. **(B, C)** Asterisks: highly significant increase in any pairwise comparison with unlabeled conditions, one-way ANOVA with the Bonferroni post hoc test, *$P \leq 0.05$, **$P \leq 0.01$, ***$P \leq 0.001$, N = 60 images from three independent experiments, error bars = SD. All unlabeled conditions (i.e., with no asterisk) were not significantly different among themselves in any pairwise comparison.

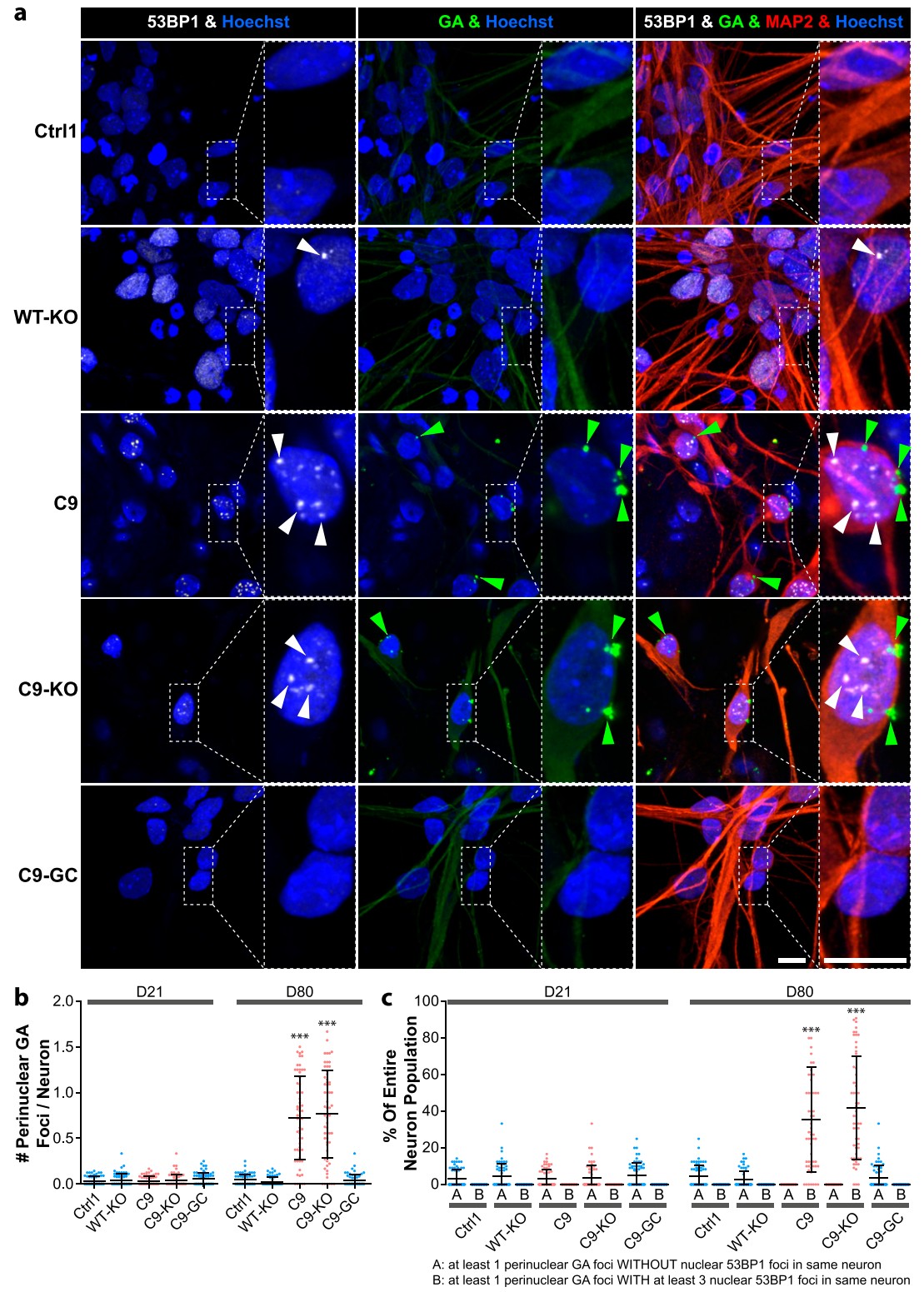

**Figure 6. DNA damage accumulation concurred with perinuclear glycine-alanine (GA) dipeptide repeat protein foci in C9ORF72 spinal MNs over ageing.**
**(A)** DNA double-strand break (DSB) marker 53BP1 (in white) in Hoechst-positive nuclei (in blue) and perinuclear GA foci (in green) in MAP2-positive neurons (in red) were revealed by confocal IF microscopy at D80 endpoints (Fig 3). Dotted boxed areas in image galleries are shown magnified on the right. Note the striking nuclear accumulation of 53BP1-positive nuclear foci (white arrowheads) in parental C9 and C9-KO that was phenocopied by WT-KO. Furthermore, perinuclear GA foci (green arrowheads) concurred with nuclear 53BP1 accumulation within the same neuron in C9 and C9-KO. Conversely, DSBs and GA foci were nearly absent in C9-GC and parental Ctrl1. Arrowheads point only to arbitrary examples. Scale bars = 10 μm. **(B)** Quantification of (A), number of perinuclear GA foci in MAP2-positive neurons at D21

essentially well after the onset of trafficking defects on D40 (Fig 8A, compare green versus red arrowhead).

### R-loops did not differ in aged C9ORF72 MNs

Next, we wished to investigate whether augmented DSBs toward endpoints (Fig 8A) were simply a late, generic epiphenomenon in the overall degenerative and apoptotic context (Fig 7) or otherwise caused by R-loops. These three-stranded DNA–RNA complexes arise during the RAN transcription of intronic HREs (Groh & Gromak, 2014; Walker et al, 2017). R-loops can pose a risk to genomic stability and cause DSBs. We performed IF stainings for R-loops over a time course (Figs 8B and S10) and found no differences in R-loop load across all lines and no change over time for any line. Thus, DSB accumulation (Fig 8A) and also axonal trafficking defects (Fig 3) were most likely not caused by R-loops.

### RNA foci increased over time in ageing C9ORF72 MNs

Next, we wished to address another aspect of RNA toxicity in C9ORF72 pathology, namely, RRE foci. These RNA accumulations arise from sense and antisense bidirectional RAN transcriptions of intronic HREs (Walker et al, 2017). We performed FISH to reveal RNA foci over the time course used previously. These experiments were performed as automated high-throughput assays on 96-well plates (Fig S11) (Rizzu et al, 2016). All HRE *C9ORF72* lines (C9, C9-1, and C9-2) exhibited strongly elevated levels of RNA foci versus their respective controls (C9-GC and Ctrl3) with both the sense (Fig 8C and E) and antisense probes (Fig 8D and F), consistent with HRE transcription. Moreover, the isogenic C9 line exhibited a steady increase in RNA foci from the beginning until D40, that is, the onset of axonal trafficking defects (Fig 8C and D, red arrowheads) to subsequently decrease hereafter. The non-isogenic C9-1 and C9-2 lines showed fairly similar kinetics (Fig 8E and F) with particularly high RNA foci levels in C9-2 revealed with the sense probe presumably because of its much higher HRE number (C9-1: >50, C9-2: ~730, Table 2). In summary, these kinetics suggest that the steady increase in RNA foci from the beginning could contribute to the onset of axonal trafficking defects on D40 in C9ORF72 MNs (Fig 3).

### Onset of GP accumulation correlated with onset of axonal trafficking defects in ageing C9ORF72 MNs

Finally, as DSBs and R-loops were not triggering axonal trafficking defects (Fig 8A and B), we favoured the augmented DPRs as causative upstream trigger (Figs 5 and 6). This view appeared plausible as DPRs were shown to inhibit microtubule-based transport directly (Fumagalli et al, 2019 *Preprint*) and to cause DNA damage through sequestration of pATM and hnRNP A3

(Walker et al, 2017; Nihei et al, 2020). As a test, we performed IF stainings for GP over a correlative time course (Figs 8G and S6). We revealed a clear sudden onset of neuritic GP foci on D40 simultaneously to the onset of axonal trafficking defects in C9 and C9-KO (Fig 8G, green and red arrowhead, Fig 3), but not in WT-KO because there were no HREs. In conclusion, the accumulation of DPRs was most likely the main driving force behind the axonal trafficking defects, possibly in concert with the steady increase in RNA foci (Fig 8C–F).

## Discussion

In this study, we wished to compare HRE *C9ORF72* phenotypically and mechanistically against mutant *FUS* and *TDP43*, all of which are common genetic aberrations causing ALS (Chia et al, 2018; Nguyen et al, 2018). We sought to combine LOF of C9ORF72 with HRE-mediated GOF in a meaningful manner with no overexpression artifacts to clarify the role of both debated mechanisms (Waite et al, 2014; Walker et al, 2017; Frick et al, 2018; Nihei et al, 2020). In contrast to FUS and TDP43 ALS, proximal in parallel to distal axonal trafficking deficits, were the hallmarks of C9ORF72 pathology in hiPSC-derived MNs along with accumulation of DPRs, RNA foci, DNA damage, and cell death. Although GOF and LOF were both contributing to the trafficking deficiencies, C9ORF72 LOF was sufficient to induce DNA damage accumulation and cell death. RAN transcription (RNA foci) and translation (DPR) were upstream of axon trafficking deficits, DSB appearance, and cell death, although SSB and DSB induction did not show similar phenotypes as C9ORF72 MNs did.

Our method of choice was fast dual-channel live imaging of axons in compartmentalized spinal MNs cultures because impaired microtubule-based organelle transport logistics are particularly vulnerable in these long neurites and often debated as a possible cause for neurodegeneration (Sheetz et al, 1998; Salinas et al, 2008; Veleri et al, 2018) as they impact diverse crucial processes such as signal progression, neurotrophic and nutritional support, target finding, neuronal plasticity and regeneration, denervation, energy support, and local deposition of mRNA. Moreover, such trafficking defects do not occur independently of other pathomechanistic events (Rothstein, 2009) such as aggregate depositions along with altered nucleo-cytosolic shuttling of disease mediators (Dormann et al, 2012; Naumann et al, 2018), suppression of neuroprotective heat shock protein induction (Tibshirani et al, 2017; Kuta et al, 2020), and impaired DNA damage response and repair (Wang et al, 2013; Rulten et al, 2014; Naumann et al, 2018). For example, we have recently shown that mutations in the nuclear location sequence of FUS hamper its nuclear import and cause its cytosolic aggregation along with its failure to recruit the nuclear DNA repair machinery to

---

versus D80, displayed as scatterplots of foci counts per neuron and image with mean (center line) and SD range (whiskers) indicated in black. For images at D21 refer to Fig S5. Note nearly absent GA foci at D21 in all lines versus drastic GA accumulation at D80 in parental C9 and C9-KO. **(C)** Quantification of (A), percentage of MAP2-positive neurons with at least one perinuclear GA focus without (A) nuclear DSBs within the same cell versus cells with both GA and at least three DSB foci (B). Scatter plots of percentages per image. Note the striking concurrence of perinuclear GA and nuclear DSB foci within same neurons at D80 in parental C9 and C9-KO as opposed to nearly absent foci of either type in all other conditions. **(B, C)** Asterisks: highly significant increase in any pairwise comparison with unlabeled conditions, one-way ANOVA with Bonferroni post hoc test, *$P \leq 0.05$, **$P \leq 0.01$, ***$P \leq 0.001$, N = 60 images from three independent experiments, error bars = SD. All unlabeled conditions (i.e., with no asterisk) were not significantly different among themselves in any pairwise comparison.

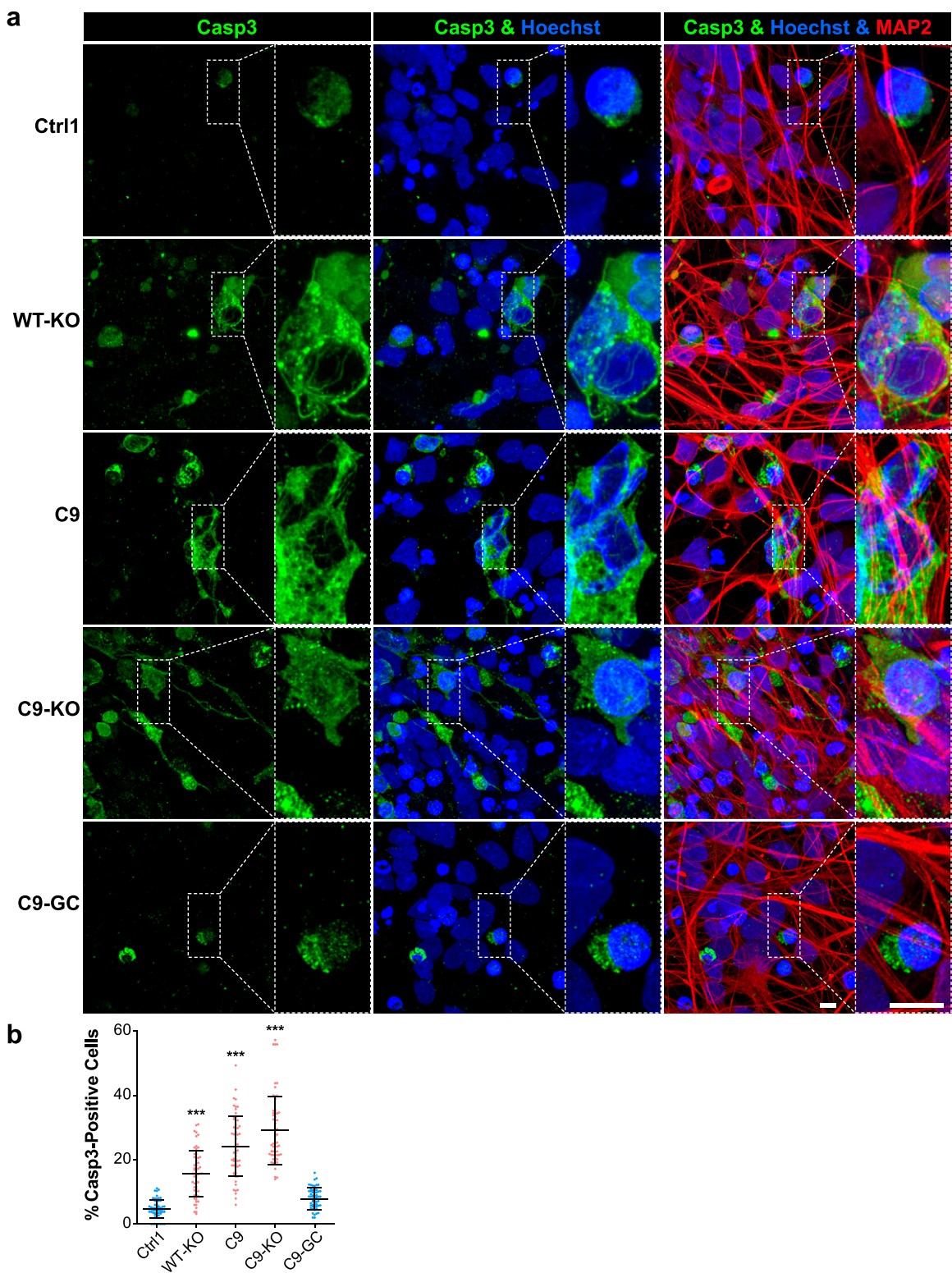

**Figure 7. Premature apoptosis occurred in C9ORF72 spinal MNs over ageing.**
**(A)** Apoptosis in MAP2-positive neurons (in red) was revealed as cytosolic rim staining around Hoechst-positive nuclei (in blue) for cleaved caspase 3 (Casp3, in green) by confocal IF microscopy at D80 endpoints (Fig 3). Dotted boxed areas in image galleries are shown magnified on the right. Note the striking accumulation of Casp3 in parental C9 and C9-KO that was phenocopied by WT-KO. Conversely, apoptosis hardly occurred in C9-GC and parental Ctrl1. Scale bars = 10 μm. **(B)** Quantification of (A), percentages of apoptotic cells in the MAP2-positive population displayed as scatterplots per image with mean (center line) and SD range (whiskers) indicated in black. Asterisks: highly significant increase in any pairwise comparison with unlabeled conditions, one-way ANOVA with the Bonferroni post hoc test, $*P \leq 0.05$, $**P \leq 0.01$, $***P \leq 0.001$, N = 60 images from three independent experiments, error bars = SD. All unlabeled conditions (i.e., with no asterisk) were not significantly different among themselves in any pairwise comparison. Remaining comparisons were WT-KO versus C9: ***, WT-KO versus C9-KO: ***, C9 versus C9-KO: ***.

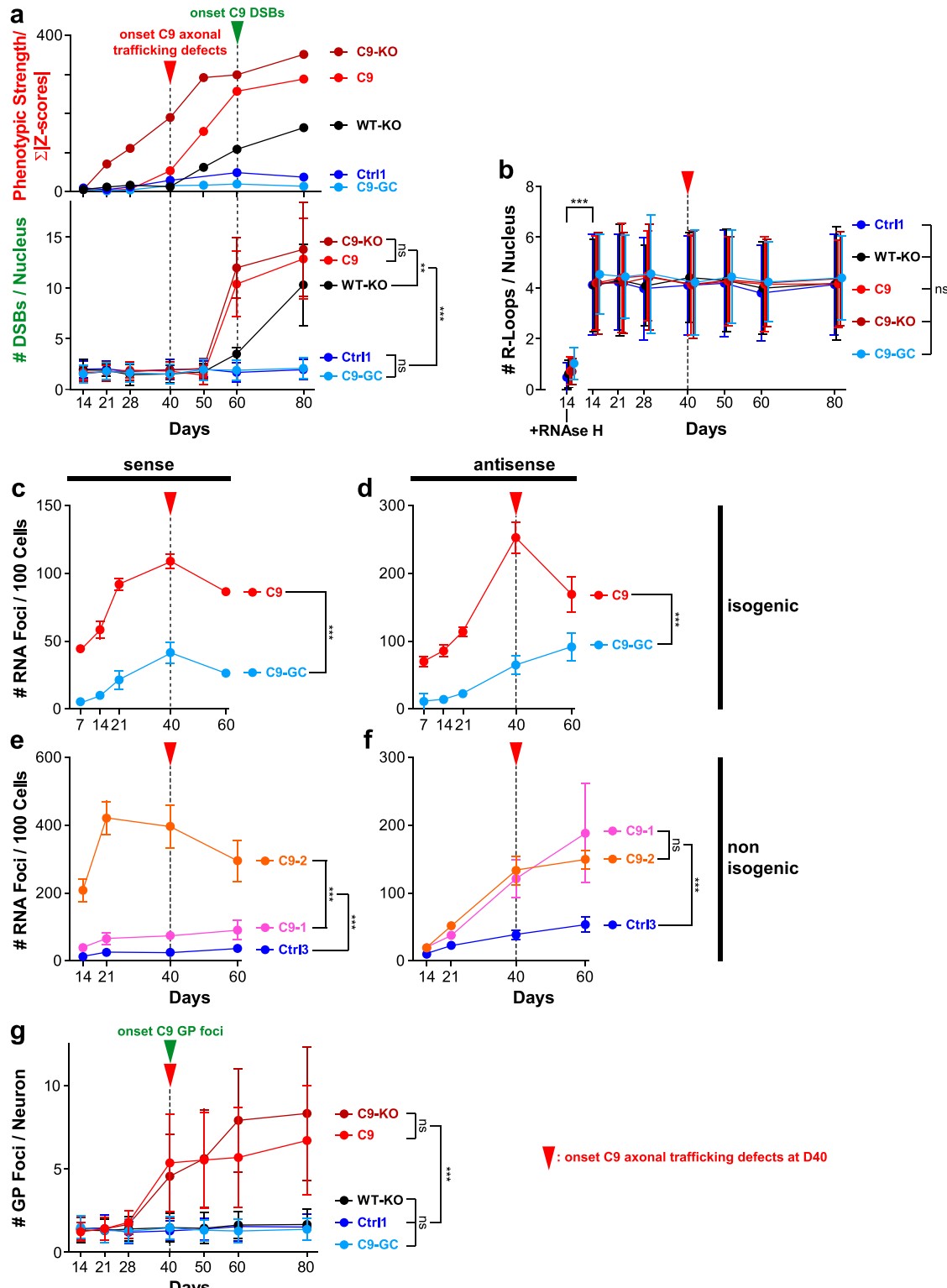

**Figure 8. Onset of axonal trafficking defects on D40 in C9ORF72 MNs correlated with the increase in glycine–proline (GP) and RNA foci, but not with DNA double-strand breaks (DSBs) and R-loops.**

**(A)** Quantification of IF stainings for DSBs with 53BP1 (lower plot) over a time course from D14 to 80 (mean DSB foci count per nucleus). Note the sudden increase in DSBs in C9 and C9-KO from D60 onward (green arrowhead), also in WT-KO to a milder extent, which occurred clearly after the onset of C9 axonal trafficking defects (read arrowhead, upper plot, phenotypic proximal strength over time as in Fig S3). (Refer to Fig S10 for corresponding image gallery.) **(B)** Likewise quantification of IF stainings for R-loops (mean foci count per nucleus). Note the same and constant count over time in all lines and the drastic drop upon RNAse H treatment on D14 to validate the

damage sites (Naumann et al, 2018). Remarkably, this impaired DNA damage response had a profound detrimental impact on distal axonal organelle motility and mitochondria activity, whereas interfering with microtubule integrity (nocodazole) or oxidative phosphorylation (oligomycin A) did cause different phenotypes (Pal et al, 2018). Thus, following the paradigm of modular cell biology (Hartwell et al, 1999), we have further refined our live imaging to obtain multiparametric HC signatures to go beyond a purely descriptive phenotyping of axonal trafficking and gain a predictive systems view of other pathomechanistic pathways as well. We have recently reported distinct signatures for mutant TDP43 and FUS (Pal et al, 2018) (Fig 1B). Given that both proteins are functionally overlapping with roles in DNA/RNA transport (Mackenzie et al, 2010), their distinct signatures demonstrated the power of our multiparametrization to resolve even subtle phenotypic differences.

As for HRE *C9ORF72*, we found very distinct axonal trafficking defects: (i) the onset of the first parameter deviations was on D40 for all lines (Fig 3A), whereas *TDP43* and *FUS* mutants showed severe axonal trafficking deficits as early as D21 (Kreiter et al, 2018; Naumann et al, 2018; Pal et al, 2018); (ii) distal and proximal axon phenotypes arose simultaneously from D40 onward, whereas all control lines exhibited only distal phenotypes (Figs 3A and S2); and (iii) HRE *C9ORF72* signatures were of distinct shape as compared with *TDP43* and *FUS* (Fig 1B), with subtle differences between lines presumably due to different genetic backgrounds and HRE repeat numbers (Figs 3A and S2). These differences pointed to a distinct spatiotemporal progression of axonal trafficking defects independent of dying back in HRE *C9ORF72* as compared with mutant *TDP43* and *FUS* and were determined by the combinatorial interplay of GOF and LOF mechanisms.

As for LOF, KO of the exonic *C9ORF72* part in wild-type control MNs (WT-KO) led to a partial mimic of the natural HRE *C9ORF72* phenotype with a premature onset of distal and delayed proximal trafficking defects (Fig 3B). The same KO in the presence of functional HREs (C9-KO) led to the most severe and earliest phenotypes, suggesting that both, GOF of HRE and LOF of C9ORF, are driving the axon trafficking deficits. Excision of HREs in *C9ORF72* (C9-GC) led to full recovery and control-like signatures (Fig 3B), identifying HREs as the primary trigger of all pathology, that is, of GOF through RAN-translated DPRs (Walker et al, 2017; Nihei et al, 2020) along with HRE-mediated LOF through reducing exonic C9ORF72 expression levels.

Next, as HREs in *C9ORF72* mediate pathology through several proposed mechanisms, we wished to further narrow down the predominant driving force behind our observed axonal trafficking defects (Fig 3). On the one hand, HRE pathology is believed to be mediated through RAN-translated DPRs (Walker et al, 2017; Nihei et al, 2020) and on the other hand, through RNA toxicity caused either by three-stranded DNA–RNA hybrid complexes (R-loops) (Groh & Gromak, 2014; Walker et al, 2017) or accumulation of RRE transcripts (RNA foci) (Walker et al, 2017), both of which resulting from bidirectional RAN transcription of intronic HREs. We performed a correlative time course analysis by IF (R-loops and DPRs, Fig 8B and G) and FISH stainings (RNA foci, Fig 8C–F) against the onset of axonal trafficking defects on D40 (Fig 3). We included IF stainings for DSBs (Fig 8A) for two reasons: (i) DSBs can arise as a consequence of DPR (Walker et al, 2017; Nihei et al, 2020) and R-loop accumulation (Groh & Gromak, 2014; Walker et al, 2017) and (ii) DSBs serve as an upstream trigger of axonal trafficking defects in FUS ALS (Naumann et al, 2018), and we wished to address this possibility for C9ORF72 ALS as well. We found a clear concurrence of a sudden onset of GP accumulation with the onset of axonal trafficking defects (Fig 8G) along with a more steady increase in RNA foci (Fig 8C–F), whereas R-loop levels remained constant over the entire time course and were indistinguishable across all lines (Fig 8B). We failed to reveal the kinetics of GA over time because of cell culture issues. However, we assume that GP served as sufficient representative for all DPRs but acknowledge the need for further studies to weight the contribution of each DPR species in more detail. Based on these results, we propose that RAN transcription with the generation of RNA foci together with RAN translation and generation of DPR seemed to be the trigger of axon trafficking deficits. DSB accumulation took place not earlier than D60 (Fig 8A), thereby suggesting that DNA damage occurred as a later consequence of DPR accumulation on D40 and RNA foci even before D40. Other feasible insults leading to DNA damage could be oxidative stress in the overall context of degenerative apoptosis. To further rule out DSBs as an upstream trigger for our observed C9ORF72 phenotypes (as in FUS ALS), we used DSB and SSB inducers to assess the impact of these treatments by HC profiling of axonal trafficking (Fig S9). We found virtually no impact for SSB and only a mimic of FUS ALS phenotypes through DSB induction (Fig S9A), that is, essentially no mimic of the proximal characteristics in aged C9ORF72 MNs. These data confirm that DSBs occurred as a late consequence, rather than the cause in C9ORF72 MN degeneration. The finding that only induction of DSBs, rather than SSBs, led to a phenocopy of FUS ALS (Fig S9A) adds an interesting mechanistic insight into the postulated nucleo-axonal cross talk (Naumann et al, 2018).

Our view of DPRs as major driving force behind the axonal transport defects and DSBs is supported by previous reports showing that they can directly inhibit microtubule-based axonal transport of mitochondria and RNA granules by interfering with

---

specificity of the stainings. (Refer to Fig S10 for corresponding image gallery.) **(C, D, E, F)** Quantification of RNA FISH images to determine the mean count of nuclear hexanucleotide repeat expansion RNA foci (RREs) per 100 cells over a time course as indicated. (Refer to Fig S11 for corresponding example images.) **(C)** RNA foci count revealed with a sense probe in the isogenic pair C9 versus C9 GC. Note the higher level of RNAi foci in parental C9 than its gene-corrected control (C9-GC) over the time course with continuous increase from the start until the onset of axonal trafficking defects (red arrowhead). **(D)** Same as (C) but with an antisense probe. **(E)** Same as (C) but for the non-isogenic trio C9-1 and C9-2 versus Ctrl3 (see Table 2). Note the increased RNAi foci count in both C9 lines as compared with Ctrl3 with C9-2 exhibiting particularly high foci counts because of its higher hexanucleotide repeat expansion repeat number (Table 2). Again, RNA foci counts increased gradually toward the onset of axonal trafficking defects in C9 (read arrowhead). **(F)** Same as (E) but with an antisense probe. **(G)** Quantification of IF stainings for neuritic GP foci in MNs from D14 to 80 (mean DSB foci count per neuron). Note the sudden increase in GP foci in C9 and C9-KO on D40 (green arrowhead) that concurred with the onset of axonal trafficking defects (red arrowhead). **(A, B, C, D, E, F, G)** Asterisks: highly significant increase in pairwise comparison as indicated, two-way ANOVA with the Bonferroni post hoc test, except RNAse H at D14 versus D14 untreated (B) that was assessed with the unpaired two-tailed *t* test. *$P \leq 0.05$, **$P \leq 0.01$, ***$P \leq 0.001$, ns, not significant, N = 60 images from three independent experiments, error bars = SD.

kinesin-1 and dynein in vivo as well as in vitro–recapitulated (i.e., cell-free) motility assays using recombinant DPRs (Fumagalli et al, 2019 Preprint). Moreover, we found augmented DSBs on D80 concurring with DPR foci within the same MN in C9 and C9-KO (Fig 6), consistent with recent reports documenting a causative role of DPRs in DSB accumulation (Walker et al, 2017; Nihei et al, 2020). Furthermore, augmented DPR and DSB foci emerged along with apoptosis (Fig 7), thereby suggesting a causative contribution of HRE-mediated pathology to neurodegeneration as reported (Walker et al, 2017; Nihei et al, 2020).

However, to our surprise, we revealed similar DSB accumulation in WT-KO MNs (Fig 8A), despite absent HREs along with apoptosis, albeit to a milder extent (Figs 7B and 8A) and an emergence of distal and proximal trafficking defects (Figs 3B and S3). This is of note because all reports pointed toward HRE-mediated DSBs (Walker et al, 2017; Nihei et al, 2020) and did not accuse LOF of C9ORF72 as the underlying cause for DNA damage accumulation. Furthermore, LOF in wild-type HRE conditions induced a distal axonal phenotype with further aggravation during ageing proceeding also to the proximal side (Figs 3B and S3). Nevertheless, the DSB is downstream from axon trafficking deficits also in the WT-KO condition because it appeared much later than axon trafficking deficits became obvious, and furthermore, SSB or DSB induction did not mimic HC WT-KO trafficking signatures (Fig S9). C9ORF72 depletion in human iPSCs was reported to lead to Arf6 activation which in turn leads to Rac1 activation, resulting in disturbance of actin dynamics in motoneurons (Sivadasan et al, 2016). Interestingly, Rho GTPases have been reported to be involved in DNA damage response, and inhibition of Rac1 is able to reduce DSBs (Wartlick et al, 2013). Thus, these might explain the observed phenotypes in C9ORF72 LOF linking them to RhoGTPAses and actin remodeling (see also Fig 4B).

Conversely, parental HRE C9ORF72 exhibited a distinct nonclassical dying back phenotype in axons but further exacerbated trafficking defects and apoptosis in C9-KO (Figs 3B and 7B). This was, however, not associated with a further exacerbation in DPR and DSB foci on D80 (Figs 5 and 6), but instead even with a 50% reduction in DPR levels in C9-KO cells by ELISA measurements as compared with parental C9, albeit at earlier time points (Abo-Rady et al, 2020). Besides, DSBs were mainly found in DPRs containing neurons (Fig 6C). Again, this finding argues against DPR-mediated DSB accumulation and apoptosis as the sole underlying cause in parental HRE C9ORF72. Other GOF mechanisms via detrimental RNA repeat expansion (RRE) foci and R-loops occurring upstream of DPR translation (Groh & Gromak, 2014; Walker et al, 2017) can feasibly explain a certain degree of DPR-independent DSB accumulation, consistent with the steady increase in RNA foci before the onset of axonal trafficking defects (Fig 8C–F). We, however, found no evidence of increased R-loops at time points where other phenotypic abnormalities were already obvious. We envision three different explanations for these conflicting data: (i) HRE-mediated GOF contributes to DSB accumulation but is not the sole cause, that is, similar to the trafficking defects (Fig 3) both GOF and LOF damage DNA concomitantly; (ii) HRE-mediated GOF is the upstream trigger and causes DSB accumulation through the LOF of exonic C9ORF72; thus, the KO of C9ORF72 in control cells leads to its LOF in the absence of HREs and consequently to DSB accumulation; and (iii) the KO of exonic C9ORF72 in control cells activates the otherwise

silent RAN translation of the normal hexanucleotide repeats (nHRs) resulting in DPR expression of distinct composition that escapes detection through available antibodies (GP and GA) as they are too short or other DPR variants accumulate and cause DNA damage. Cases (i) and (ii) raise the question of how a LOF of exonic C9ORF72 can cause DSBs. Because C9ORF72 has documented roles in endocytosis with its RNAi-mediated KD impacting on lysosomal degradation and autophagy (Farg et al, 2014), its LOF in ALS could stress cells through perturbed trafficking, protein turnover, and ROS accumulation (Ferraiuolo et al, 2011; Sasaki, 2011; Otomo et al, 2012), eventually leading to DSB accumulation as neurons are dying. Case (iii) raises the question of how the KO of exonic C9ORF72 can activate RAN translation of the fewer intronic nHRs in the wild type. Several reports have highlighted the impact of HRE-mediated RAN translation on C9ORF72 expression (Waite et al, 2014; Sivadasan et al, 2016; Frick et al, 2018), but this down-regulation is unlikely to be a one-way effect because KO of the exonic C9ORF72 in parental HRE cells reduced vice versa HRE-mediated GP expression (Abo-Rady et al, 2020) by 50%, thereby indicating a mutual impact of both the intronic HRE and exonic C9ORF72 parts in the RNA transcript on their respective counterpart translation. Because the KO of C9ORF72 was realized by excision of the start codon, an alternative translation start codon further inwards along with a premature stop codon could feasibly lead to a nonsense-mediated decay of the RNA transcript, thereby limiting HRE-mediated DPR expression, whereas in the WT-KO, the lack of HREs could instead confer enhanced RNA stability, thereby activating RAN translation of nHRs or at least augmenting RRE foci and R-loops sufficient to increase DSBs (Walker et al, 2017). This explanation appears feasible as overexpression of DPR constructs of quasi–wild-type length was sufficient to produce RRE foci and R-loops, albeit to a lesser extent (Walker et al, 2017). Consistently, treatment of iPSC-derived control MNs with recombinant DPRs of only 20 repeats was shown to be sufficient to inhibit the microtubule-based motor proteins kinesin-1 and dynein in axons (Fumagalli et al, 2019 Preprint).

In conclusion, spatiotemporal disease progression of axonal organelle trafficking was distinct in HRE C9ORF72 as compared with FUS and TDP43 (Kreiter et al, 2018; Naumann et al, 2018; Pal et al, 2018) because of concomitant GOF and LOF mechanisms causing trafficking defects along with accumulation of DPRs, RNA foci, and DSBs. Many of the reported GOF mechanisms ranging from RNA foci, erratic transcription and splicing (Walker et al, 2017; Nihei et al, 2020), R-loops (Groh & Gromak, 2014; Walker et al, 2017), DPR-mediated DSBs (Walker et al, 2017; Nihei et al, 2020), and axonal trafficking defects (Fumagalli et al, 2019 Preprint) had so far unclear contributions to the overall pathology. Our data point clearly to C9ORF72 LOF together with DPRs along with RNA foci as major contributors of axon trafficking deficits and later cell death, whereas R-loops did not show any differences, and DNA damage seemed to be a downstream effect, rather than a cause of phenotypes. Conversely, the necessity of concomitant LOF highlights a novel aspect calling to investigate the underlying mechanism that presumably comprises the known roles of C9ORF72 in endocytosis and autophagy (Otomo et al, 2012; Farg et al, 2014; Shi et al, 2018), consistent with enrichment of LOF-specific DEGs in a functional PPI cluster of endocytosis revealed by our transcriptomic analysis of C9 versus C9-KO (Fig 4D). Furthermore, our PPI network mapping (Fig 4)

did not enable a dissection of the GOF–LOF interplay down to molecular players but revealed functional clusters that suggest alterations in cytoskeletal interactions and energy supply as a systemic cause further upstream that impairs microtubule-dependent trafficking of several organelle types through a similar mechanism. In line with this view is the gross similarity of the lysosomal versus mitochondrial parts in our HC profiles (Fig 3), especially toward the endpoints. Consistently, when we systemically disrupted all microtube-dependent organelle trafficking with nocodazole (Pal et al, 2018), our HC profiling revealed very similar alterations in both the mitochondrial and lysosomal parts, as shown in Fig 2 of that report (Pal et al, 2018). Ditto when we inhibited mitochondrial ATP production with oligomycin A as the resultant energy deprivation systemically stalled motor proteins on all organelle types (Pal et al, 2018). Interestingly, most of the LOF-specific DEGs were shared with HRE-mediated GOF DEGs (Fig 4A), suggesting that most LOF insults are either a consequence of upstream HRE RAN transcription/translation or LOF is the strongest driver of pathology because homozygous KO even fueled similar DEGs. The latter would also fit to the data that C9-KO has the strongest phenotype although lacking earlier appearance or higher levels of DPRs (Fig 8). This finding is in line with the known reduction of exonic C9ORF72 gene expression through HREs (Waite et al, 2014; Sivadasan et al, 2016; Frick et al, 2018).

Finally, TDP43 did not exhibit any alteration in its prominent nuclear localization even when the most severe trafficking and DSB phenotypes occurred, thereby arguing against an upstream role in the interplay of GOF and LOF mechanisms or otherwise pointing to limitations of our cell model. Nevertheless, our rescue data on C9-GC MNs clearly indicate the therapeutic value of intervening specifically against the HREs in the C9ORF72 locus. Such approaches using antisense oligonucleotides (ASOs) selectively targeting HRE transcripts already exist and yielded promising results (Jiang et al, 2016). ASOs are certainly effective in reducing DPR expression, and further studies should be performed to clarify whether this interference is sufficient for clinical application as other GOF mechanisms stemming from RRE foci and R-loops (Walker et al, 2017) are likely to persist. Moreover, our LOF data (WT-KO) of partial HRE C9ORF72 mimic clearly emphasize the need of further refinement to ensure clinical ASO applications leave C9ORF72 expression unaltered as the compensation through the healthy allele appears to be insufficient. Clearly, the critical role of the C9ORF72 LOF was unveiled through our homozygous KO models, thereby validating the power of this tool for mechanistic dissections, although it is potentially less faithful in mimicking the ALS clinic. One drawback of our KO models to address LOF of exonic C9ORF72 might be the fact that we generated a homozygous loss of function. This genetic modification was realized by CRISPR/Cas9n–mediated excision of the ATG start codon in exon 2 of the C9ORF72 gene locus, leading to deactivated translation of both alleles with no remaining C9ORF72 expression (Abo-Rady et al, 2020). This strategy, unlike conventional KOs, enabled us to eliminate C9ORF72 translation without compromising RAN translation from intron 1. Technically, this strategy is unsuitable to target only one allele; therefore, only homozygous KO lines were obtained. On the first glance, homozygosity appears unfavourable as most ALS cases are heterozygous. However, conceptually we disfavoured a heterozygous line, albeit closer to the ALS clinic because the purpose of our KO lines was not

to clinically mimic the LOF of C9ORF72 by carefully fine-tuning its levels moderately down to the levels seen in clinical patients. Rather, we wished to clarify whether there is an LOF contribution to the overall pathology per se. Therefore, we sought to unmask the LOF in the complex, potentially obscuring interplay with other multiple GOF mechanisms at maximum clarity. In essence, we propose to further boost the efficacy of ASOs in eliminating HRE-mediated DPR expression by co-targeting RPS25, a small ribosomal protein subunit of 25 kDa required for RAN translation of DPRs that was recently identified in a genetic screen (Yamada et al, 2019).

# Materials and Methods

### Generation, gene editing, and differentiation of human iPSC lines to MNs

All procedures were performed in accordance with the Helsinki convention and approved by the Ethical Committee of the Technische Universität Dresden (EK45022009 and EK393122012). Patients and controls gave their written informed consent before skin biopsy. The generation and expansion of iPSC lines from healthy control and familiar ALS patients with defined mutations in the FUS or TDP43 gene and HREs in C9ORF72 (Table 2) were recently described (Donnelly et al, 2013; Sivadasan et al, 2016; Higelin et al, 2018; Kreiter et al, 2018; Naumann et al, 2018; Catanese et al, 2019). In brief, following the skin biopsy, reprogramming with Oct4, Sox2, Klf4, and c-Myc expression, and colony selection, iPSCs were cultured in TeSR-E8 medium (Stem Cell Technologies) at 37°C and 5% $CO_2$ with daily media changes and regular passaging. For gene targeting, plasmids containing the Cas9n and sgRNAs were transfected using FuGENEHD (Promega) followed by selection. Clonal lines were expanded and characterized. Isogenic C9-KO and C9-GC lines from parental C9 (Table 2) were generated by CRISPR/Cas9n–mediated gene editing and fully characterized in the Sterneckert laboratory (Abo-Rady et al, 2020). The subsequent differentiation to neuronal progenitor cells (NPCs) and further maturation to spinal motor neurons (MNs) were performed, as described (Abo-Rady et al, 2020). In brief, iPSCs were seeded on mouse embryonic fibroblasts and cultured in human ES medium supplemented with 5 ng/ml FGF2 and 2.5 ng/ml activin A (ActA; eBioscience). On day 3, supplements were removed, cells detached with collagenase (Life Technologies), and neuronal induction initiated with hES medium supplemented with 200 $\mu$M ascorbic acid (AA; Sigma-Aldrich), 3 $\mu$M CHIR 99022 (Axon Medchem), 10 $\mu$M SB 431542 (Biomol), 5 $\mu$M dorsomorphin (DM; Absource), and 5 $\mu$M ROCK inhibitor (Y-27632; Abcam), resulting in the formation of embryoid bodies (EBs). The medium was changed every other day, and after 4 d, the medium was changed to N2B27 medium——DMEM F12 medium 1:1 neurobasal medium supplemented with N2 and B27 (all Thermo Fischer Scientific), penicillin, streptomycin, glutamine (MerckMillipore)—and supplemented with 200 $\mu$M AA, 3 $\mu$M CHIR, 0.5 $\mu$M valproic acid (VPA; Biomol), 0.5 $\mu$M purmorphamine (PMA; Santa Cruz Biotechnology), 0.5 $\mu$M DM, 10 $\mu$M SB 431542, and 0.1 $\mu$M retinoic acid (RA; Sigma-Aldrich). After two more days, EBs were transferred to a 12-well (Corning) covered dish, dissociated, and grown in monolayer as NPCs. NPCs were kept in culture and split with Accutase (Sigma-Aldrich). To induce differentiation, N2B27 medium was supplemented with 200 $\mu$M AA,

3 $\mu$M CHIR, 0.5 PMA, 1 $\mu$M RA, 10 ng/ml BDNF, and 20 ng/ml GDNF (both PeproTech). After 6 d, the medium was changed to maturation medium consisting of N2B27 with 200 $\mu$M AA, 10 ng/ml BDNF, 20 ng/ml GDNF, 200 $\mu$M dibutyryl cyclic adenosine monophosphate (dbcAMP; Selleck Chemicals), 1 ng/ml TGF$\beta$3 (PeproTech), and 5 $\mu$M DAPT (Biomol). 2 d later, MNs in the early maturation phase were finally reseeded into MFCs. After 1 wk, DAPT was removed from the medium, and MNs were kept in culture until D80. (For final maturation and ageing, see the following paragraph.)

### Final maturation and ageing of MNs in MFCs

The coating and assembly of MFCs (Xona) to prepare for the seeding of MNs was performed as described (Naumann et al, 2018). In brief, 3.5-cm glass-bottom dishes (Nunc) were coated with poly-L-ornithine (P4957, 0.01% stock diluted 1:3 in PBS; Sigma-Aldrich) overnight at 37°C. After three washing steps with sterile water, they were kept under the sterile hood for air-drying. MFCs were sterilized with 70% ethanol and left for drying. Next, the MFCs were dropped onto the dishes and carefully pressed on the glass surface for firm adherence. The system was then perfused with laminin (11243217001, 0.5 mg/ml stock diluted 1:50 in PBS; Roche) for 3 h at 37°C. For seeding MNs, the system was once washed with the medium, and then 10 $\mu$l containing a high concentration of cells (3 × 10$^7$ cells/ml) were directly injected into the main channel connecting two wells. After allowing for cell attachment over 30–60 min in the incubator, the still empty wells were filled up with the maturation medium. 2 d after seeding, the medium was replaced in a manner which gave the neurons a guidance cue for growing through the microchannels. Specifically, a growth factor gradient was established by adding 100 $\mu$l N2B27 with 500 $\mu$M dbcAMP only to the proximal seeding site and full maturation medium to the distal exit site. The medium was replaced in this manner every third day. As MNs were seeded into one site (Fig 1A) of MFCs only, fully compartmentalized cultures were obtained with proximal somata and their dendrites being physically separated from their distal axons as only the latter type of neurites was capable to grow from the proximal seeding site through a microgroove barrier of 900-$\mu$m-long microchannels to the distal site. Subsequent imaging in MFCs (Fig 1A) was performed on D14, 21, 28, 40, 50, 60, and 80 of axon growth and MN maturation (D0 = day of seeding into MFCs). The purity of our spinal MN cultures was assessed in several of our previous publications (Naumann et al, 2018; Abo-Rady et al, 2020). In essence, in the proximal seeding chamber, our differentiation protocol yielded up to 80% mature MNs and a remaining mix of neuronal progenitors and other neuronal subtypes. The cultures were devoid of glia cells and astrocytes. Neurites penetrating the microchannels and sprouting out at the distal exit are virtually 100% axon-pure (Gla$\beta$ et al, 2020).

### Live imaging of MN in MFCs

Movie acquisition at strictly standardized readout windows at the distal exit and the proximal entry of the MFC microchannels (Fig 1A) was performed as described (Naumann et al, 2018; Pal et al, 2018). In brief, to track lysosomes and mitochondria, cells were double-stained with live cell dyes LysoTracker Red DND-99 (Cat. No. L-7528; Molecular Probes) and MitoTracker Deep Red FM (Cat. No. M22426; Molecular

Probes) at 50 nM each. Trackers were added directly to culture supernatants and incubated for 1 h at 37°C. Live imaging was then performed without further washing off cells in the Center for Molecular and Cellular Bioengineering, Technische Universität Dresden (CMCB) Light Microscopy Facility with a Leica HC PL APO 100× 1.46 oil-immersion objective using an inverted fluorescent Leica DMI6000 microscope enclosed in an incubator chamber (37°C, 5% CO$_2$, humid air) and fitted with a 12-bit Andor iXON 897 EMCCD camera (512 × 512 pixel, 16 $\mu$m/pixels on chip, 229.55 nm/pixel at 100× magnification with intermediate 0.7× demagnification in the optical path through the C-mount adapter connecting the camera with the microscope). (For more details, refer to https://www.biodip.de/wiki/Bioz06_-_Leica_AFLX6000_TIRF and our previous publication [Naumann et al, 2018].) Fast dual-color movies were recorded at 3.3 frames per second per channel over 2 min (400 frames in total per channel) with 115-ms exposure time as follows: LysoTracker Red (excitation: 561 nm laser line, emission filter TRITC 605/65 nm) and MitoTracker Deep Red (excitation: 633 nm laser line, emission filter Cy5 720/60 nm). Dual-channel imaging was achieved sequentially by fast switching between both laser lines and emission filters using a motorized filter wheel to eliminate any cross talk between the two trackers.

Our live setup (at 100× magnification and using the Andor camera as described earlier) covers in its viewing field at each readout position 2 channels in parallel, each with 117.53 $\mu$m of their entire length (900 $\mu$m) from either the distal exit or from the proximal entry.

### Phenotypic HC profiling

We recently published a comprehensive description of the whole automated analytical pipeline starting from object recognition in raw movie data to final multiparametric signature assembly (Pal et al, 2018). In brief, organelle recognition and tracking were performed with FIJI TrackMate plugin and organelle shape analysis with our custom-tailored FIJI Morphology macro. Both tools returned a set of nine master parameters in total for each organelle type (Mito versus LysoTracker) and readout position (i.e., distal versus proximal), for example, mean speed and diameter. Subsequent data mining of individual per-movie result files was performed in KNIME to assemble complete final result files with annotated per-organelle parameters, thereby allowing to pool all data of each experimental condition (e.g., all data for MitoTracker at the distal readout position on D21 for cell line Ctrl1). In addition, two post-processing parameters were calculated in KNIME, that is, the track ratio anterograde/retrograde movement (Fig 1B, parameter 9), as described (Pal et al, 2018), and the percentage of moving tracks (Fig 1B, parameter 10), defined as the percentage of tracks with a minimum track displacement of 1.2 $\mu$m as an arbitrary threshold for moving organelles as opposed to stationary ones. Therefore, a total number of 11 master parameters were finally obtained for each organelle type and readout position. Z-scores were calculated for each parameter (Pal et al, 2018) to express its deviation from pooled control lines at the proximal readout and assembled to whole HC signatures comprising a total of 44 parameters (Fig 1B) as the set of 11 master parameters was applied four times (i.e., distal versus proximal and Mito versus LysoTracker, Fig 1B). Clustering of whole signatures (Fig 1C and D) was performed with the KNIME node "Hierarchical Clustering" as described (Pal et al, 2018).

## Bulk statistics, replicas, and blinding

We imaged distal versus proximal axons in MFC microchannels at strictly standardized positions, as highlighted in the cartoon of Fig 1A. As somata are always outside the viewing fields, it was not possible to assign the imaged axons to individual soma; therefore, the precise cell number in our experiments was unknown. However, we always imaged two channels in parallel per movie and acquired a minimum of five movies (10 microchannels) per MFC (one technical replica). Therefore, the theoretical minimum number of cells was 10 per MFC, if each microchannel had only been occupied by one axon. However, we always observed axon bundles (Fig 2A and Videos 1 and 2) of 5–30 axons per microchannel, that is, in total 50–300 axons per imaged MFC. Given the number of technical replicas (three) and experiments (at least three, often five), we often imaged thousands of axons per condition in our pooled data sets. As a single microchannel yielded typically already hundreds of individual organelle tracks per movie, our final bulk statistics were very robust and significant. Furthermore, we eliminated the risk of biasing on three levels: (i) the operator at the microscope always imaged cell culture samples labeled with a blinding code only, that is, was unaware of the actual genotype; (ii) there was hardly any freedom in the choice of the viewing field at our strictly standardized readout positions at the microscope (Fig 1A); thus, it was virtually impossible to introduce any bias during the movie acquisition; and (iii) the acquired raw movies were passed to a different person who performed the strictly standardized tracking analysis that was largely automated with no freedom to introduce any bias (Pal et al, 2018), and no subset analysis with a different setting of the object recognition and tracking algorithm was performed.

## Immunofluorescence stainings

For immunofluorescence staining, cells were washed twice with PBS without $Ca^{2+}/Mg^{2+}$ (Life Technologies) and fixed with 4% PFA in PBS for 10 min at room temperature, except for stainings for R-loops, which required fixation with pure methanol for 30 min on ice. PFA/methanol was aspirated off, and cells were washed three times with PBS at room temperature. Cells were then permeabilized for 10 min in 0.1% Triton X solution and subsequently incubated for 1 h at RT in blocking solution (1% BSA, 5% donkey serum, 0.3 M glycine, and 0.02% Triton X in PBS). Following blocking, primary antibodies were diluted in blocking solution, and cells were incubated with primary antibody solution overnight at 4°C, except for the γH2A.X antibody, which was kept for only 2 h at room temperature on the fixed material. The following primary antibodies were used: mouse anti-γH2A.X (1:500, #05–636; Millipore), rabbit anti-53BP1 (1:1,000, NB100-304; Novus Biologicals), chicken anti-MAP2 (1:1,000, ab5392; Abcam), rabbit anti-Casp3 (1:1,000, #9661; Cell Signaling Technology), mouse anti R-loop (DNA–RNA hybrid clone S9.6, 1:250, Kerafast ENH001), rat anti-GP (1:500, clone 18H8), and mouse anti-GA (1:500, clone IAI2) (the latter two were generously provided by Dieter Edbauer [Nihei et al, 2020]). Nuclei were counterstained using Hoechst (Life Technologies). To confirm the specificity of the anti–R-loop stainings, MNs fixed with methanol and permeabilized (see the aforementioned text) were incubated with RNase H enzyme (Merck 10786357001, final 100 U/ml) in 3% BSA in PBS overnight at 4°C before proceeding with the aforementioned IF staining protocol, that is, the blocking step.

## Treatments of spinal MNs with RNase H and DNA topoisomerase inhibitors

Etoposide (E1383; Merck) was dissolved in DMSO to obtain a 5 mM stock. BNS-22 (614853; Merck) was dissolved in DMSO to obtain a 10 mM stock. Camptothecin (C9911; Merck) was dissolved in DMSO to obtain a 10 mM stock. Final work concentrations were 5 $\mu$M for etoposide and 10 $\mu$M for BNS-22 and camptothecin. Each inhibitor was added exclusively to the proximal soma site of MFCs 72 h before assays.

## Image quantification and statistics

For IF microscopy on fixed cells, a minimum of three independent experiments based on three distinct differentiation pipelines was always performed. 20 images per experiment of mature MNs at defined time points after seeding into MFCs were examined, and the numbers per neuron of foci representing R-loops, DSBs, DPRs, or TDP43 in MAP2-positive masks were determined using the particle analyser of FIJI after thresholding with the triangle background subtraction algorithm. Casp3-positive cells and TDP43-positive nuclei were counted manually. Statistical analysis was performed using GraphPad Prism version 5.0. If not otherwise stated, one-way ANOVA was used for all experiments with the Bonferroni post hoc test to determine statistical differences in pairwise comparisons. $*P < 0.05$, $**P < 0.01$, $***P < 0.001$, $****P < 0.0001$ were considered significant. Data values represent mean ± SD unless indicated otherwise.

For HC phenotypic profiling, data of at least four independent experiments based on four different differentiation pipelines were pooled to calculate Z-scores. We verified that the inter-experimental variability was marginal as compared with the interline variability, as described (Pal et al, 2018), thereby validating the pooling of data across all experiments.

## Identification and functional clustering of DEGs from high-throughput RNA-seq data sets

We obtained the RNA-seq data set GSE143743 (Abo-Rady et al, 2020) (spinal motor neurons) of C9ORF72-ALS patients from the National Center for Biotechnology Information (NCBI) Gene Expression Omnibus database (Barrett et al, 2013). The GSE143743 data set containing nine samples from an ALS patient with *C9ORF72* mutation versus isogenic controls (C9, C9-GC, and C9-KO, Table 2) was obtained from iPSC-derived MNs. From this data set, we analyzed a subset comprising six data sets each for C9-GC and C9-KO (GSM4273606, GSM4273607, GSM4273608, GSM4273609, GSM4273610, and GSM4273611) compared with three parental C9 data sets (GSM4273603, GSM4273604, and GSM4273605) that met the inclusion requirements, respectively.

The RNA-seq data (GSE143743) were analyzed via Partek Genomics Suite 7.0 software (Partek Inc) under the "RNA seq" workflow menu according to the standard pipeline method. Sequencing reads (BAM files) were processed and mapped for mRNA annotations on the basis of hg38_ensembl_release100. Data were normalized and expressed as reads per kilobase per million reads (RPKM). The expression levels of genes in each sample and the corresponding fold changes were estimated by DESeq2 1.16.1 (Partek Inc) (Love et al, 2014). To generate significantly DEGs among different samples, a cutoff of the false discovery rate adjusted $P < 0.05$ (Benjamini–Hochberg correction) and fold change FC ≥ 1.5 were applied. The lists of DEGs obtained in this

study were considered for PPI analysis (see the following method) using STRING database. The Venn diagram depicting intersections of DEGs analysis was made to categorize the data into two groups of different expression patterns using public software "An Interactive Tool for Comparing Lists with Venn's Diagrams (2007)" available online: https://bioinfogp.cnb.csic.es/tools/venny/index.html.

### PPI network analysis

To better understand the functional interactions of the DEGs and identify the best candidate genes in C9ORF72-ALS disease subtype, a comprehensive PPI network of their encoding products was constructed and analyzed by using the Search Tool for the Retrieval of Interacting Genes/ Proteins (STRING, version 11.0, https://string-db.org/) (Szklarczyk et al, 2019). The STRING database collects and integrates all functional associations between the genes/proteins by consolidating known and predicted interaction data derived from sources including database, experimental, co-expression, text mining, co-occurrence, neighborhood, and gene fusion with the highest confidence score. The statistical enrichment analysis in STRING indicated that the PPI interactomes were significantly enriched ($P$-value < 0.05), and it has been well documented that clustering algorithms were useful for grouping proteins into functional clusters or modules. Therefore, functional clusters were identified using the Markov clustering algorithm (MCL) (Brohée & van Helden, 2006) provided by the STRING database. The "inflation" parameter that defines the precision of the clustering between regions of strong and weak interactions was set to 1.5 (the higher the inflation, the more clusters you obtain with weak interactions). In brief, the complete list of DEGs (both up- and down-regulated) was imported into STRING v.11.0, and the PPI network was built independently for each data set with the highest level of confidence between interactions (highest score of >0.9). The resulting interactome map was further divided into subnetworks of proteins, each of which represented potential functional clusters or functional modules (using MCL clustering). In the network, genes/proteins represent nodes and connecting lines represent interactions between nodes. The "degree" of a node is indicated by the number of connecting lines it has to other nodes, that is, the higher the degree of a node (hub proteins/genes), the more important its predicted biological function. A node degree ≥15 was used to identify highly interactive hub networks of functional and biological relevance. For a larger number of seed proteins, the "zero order network" construction was performed through the NetworkAnalyst tool (Xia et al, 2015) to contain only the original seed proteins that directly interact with each other, preventing the well-known "hairballeffect" and allowing for better visualization and interpretation. Two different kinds of interactome

maps were built: either (i) only direct interactions within the original set of DEGs were allowed (Fig 8B and D) or (ii) by adding a maximum of 100 additional binding partners ranked by interaction score (see previous text) to the original DEGs. Of these, a maximum of 50 were allowed to interact directly with the entire set of original DEGs as "first shell," and a further maximum of 50 "second shell" partners were allowed to connect to the entire "first shell" as indirect binding partners to the original DEGs (Fig 8C, E, and F). Assuming that not all nodes of a functional module were detectable as DEGs, the addition of further interactors predicted missing nodes, thereby completing the modules and improving their visibility.

### RNA FISH

The RNA FISH to detect RNA foci in *C9ORF72* hexanucleotide repeat (GGGGCC)n expansion carrier samples was carried out as previously described (Rizzu et al, 2016). In brief, MNs were seeded at 50,000 cells per well in 96-well microtiter plates (6055302; Perkin Elmer). Cells were fixed in 4% PFA for 15 min and washed twice in PBS for 5 min each. The cell membrane was permeabilized by treatment with 0.2% Triton/PBS for 10 min, washed twice in PBS for 5 min, and dehydrated twice in 70% ethanol for 2 min, once in 100% ethanol for 2 min and finally air-dried. In a chemical fume hood, 100 $\mu$l of hybridization solution (without probe) was added to each well for 20 min at 66°C. The locked nucleic acid (LNA) nucleotides probes used were purchased from Qiagen (product number 339501) for sense (CCCCGG)$_{25}$ (Lot: 274175518) and antisense sequence (GGGGCC)$_{25}$ (Lot: 274175516). The composition of hybridization mixture is provided in the table. LNA nucleotides probes were thawed on ice and diluted in hybridization buffer to a final concentration of 40 nM and vortexed for 30 s. The probes were then denatured at 80°C for 5 min on a heat block and chilled on ice. The hybridization solution was aspirated off from cells and 100 $\mu$l of denatured probe work solution added per well. Plates with lids were sealed with aluminium foil and incubated in the hybridization oven at 66°C for 2 h. Subsequent washing steps were as follows: wash 1: (2× SSC/0.1% Tween 20 RNase-free) once for 5 min, at RT; wash 2: three times in the same solution for 10 min at 65°C. This step removed nonspecific and repetitive DNA/ RNA hybridization. Hoechst 33342 stock solution (10 mg/ml) diluted 1: 10,000 in RNase-free $H_2O$ was then added for 10 min at RT, protected from light. Cells were finally washed twice with DEPC water for 5 min and stored in 0.2 × SCC buffer for imaging.

### FISH image acquisition and analysis

The fluorescence images were acquired on an automated microscope (Yokogawa CV7000) at predetermined settings using a water

| Reagent | Stock concentration | Work concentration | Volume per 10 ml |
|---|---|---|---|
| Dextran sulphate | 50% | 10% | 2 ml |
| Ribonucleoside–vanadyl complex | 200 mM | 10 mM | 0.5 ml |
| SSC buffer | 20 fold | 2 fold | 1 ml |
| Sodium phosphate buffer pH 7 | 1 M | 50 mM | 0.5 ml |
| Formamide | 100% | 50% | 5 ml |
| DEPC-treated water | n/a | n/a | 1 ml |
| | | Total = 10 ml | |

immersion 60× objective, and the images were analyzed in Columbus (Perkin Elmer). The results are represented as RNA foci count normalized to 100 cells or as percentage of cells positive for foci.

## Data Availability

All data and materials are available upon request to A Hermann. All data not included in the main manuscript or appendix can be provided through deposition on a public server to be determined with the editor.

## Supplementary Information

## Acknowledgements

We acknowledge the great help in cell culture by Anett Böhme, Sylvia Kanzler, Andrea Kempe, and Katja Zoschke. The Light Microscopy Facility of the Center for Molecular and Cellular Bioengineering, Technische Universität Dresden (CMCB) provided excellent support for all live imaging experiments. We thank Ronny Sczech for programming the original FIJI/KNIME analytical HC organelle trafficking pipeline. We thank Dieter Edbauer for generously providing GA and GP antibodies. Funding: This work was supported, in part, by the Else Kröner foundation to M Naumann, "Deutsche Gesellschaft für Muskelerkrankungen (He2/2)" to A Hermann, the NOMIS foundation to A Hermann, the Helmholtz Virtual Institute "RNA dysmetabolism in ALS and FTD (VH-VI-510)" to A Hermann, an unrestricted grant by a family of a deceased ALS patient to A Hermann, and the Stiftung zur Förderung der Hochschulmedizin in Dresden. A Hermann is supported by the Hermann und Lilly Schilling-Stiftung für medizinische Forschung im Stifterverband. R Günther is supported by NiemALS Aufgeben e.V.

### Author Contributions

A Pal: conceptualization, data curation, formal analysis, investigation, validation, visualization, methodology, and writing—original draft, review, and editing.
B Kretner: data curation, formal analysis, investigation, and methodology.
M Abo-Rady: resources, data curation, formal analysis, and investigation.
H Glaβ: resources, data curation, software, formal analysis, investigation, visualization, and methodology.
BP Dash: data curation, formal analysis, investigation, visualization, and methodology.
M Naumann: resources, data curation, and methodology.
J Japtok: resources, data curation, formal analysis, investigation, and methodology.
N Kreiter: resources, data curation, formal analysis, and investigation.
A Dhingra: data curation, formal analysis, investigation, visualization, and methodology.
P Heutink: data curation, formal analysis, supervision, investigation, and methodology.
TM Böckers: resources, data curation, and formal analysis.
R Günther: data curation, formal analysis, supervision, funding acquisition, investigation, and methodology.
J Sterneckert: resources, data curation, investigation, and methodology.
A Hermann: conceptualization, resources, supervision, funding acquisition, validation, investigation, project administration, and writing—original draft, review, and editing.

### Conflict of Interest Statement

The authors declare that they have no conflict of interest.

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
