## [Reviewer comments · Life Science Alliance]

Life Science Alliance

Concomitant gain and loss of function pathomechanisms in C9ORF72 amyotrophic lateral sclerosis

Arun Pal, Benedikt Kretner, Masin Abo-Rady, Hannes Glaß, Banaja Dash, Marcel Naumann, Julia Japtok, Nicole Kreiter, Ashutosh Dhingra, Peter Heutink, Tobias Böckers, Rene Günther, Jared Sternecker, and Andreas Hermann

DOI: <https://doi.org/10.26508/lsa.202000764>

Corresponding author(s): Andreas Hermann, University of Rostock

Review Timeline:

Submission Date:	2020-05-04
Editorial Decision:	2020-05-29
Revision Received:	2020-12-14
Editorial Decision:	2021-01-19
Revision Received:	2021-01-26
Accepted:	2021-01-27

Scientific Editor: Shachi Bhatt

Transaction Report:

May 29, 2020

Re: Life Science Alliance manuscript #LSA-2020-00764-T

Andreas Hermann
University of Rostock

Dear Dr. Hermann,

Thank you for submitting your manuscript entitled "Concomitant gain and loss of function pathomechanisms in C9ORF72 amyotrophic lateral sclerosis" to Life Science Alliance. The manuscript was assessed by expert reviewers, whose comments are appended to this letter.

As you will see, the reviewers appreciate your analyses and they provide constructive input on how to further strengthen it. We would thus like to invite you to submit a revised version of your manuscript to us, addressing the individual points raised. We realize that it may be difficult to link the effects on motility to the effects on DSBs and apoptosis and to add more mechanistic insight, so these concerns do not need to get fully addressed.

The typical timeframe for revisions is three months. We are aware that the current COVID-19/Sars-CoV-2 pandemic affects many laboratories, and we have adapted our guidelines accordingly to extend the revision time if needed. Please note that papers are generally considered through only one revision cycle, so strong support from the referees on the revised version is needed for acceptance.

Thank you for this interesting contribution to Life Science Alliance. We are looking forward to receiving your revised manuscript.

Sincerely,

Reilly Lorenz
Editorial Office Life Science Alliance
Meyerhofstr. 1
69117 Heidelberg, Germany
t +49 6221 8891 414
e contact@life-science-alliance.org
www.life-science-alliance.org

B. MANUSCRIPT ORGANIZATION AND FORMATTING:

Reviewer #2 (Comments to the Authors (Required)):

In this manuscript, Pal et al. summarized a potential mechanism by which the C9orf72 hexanucleotide GGGGCC repeat expansion (HRE) induces amyotrophic lateral sclerosis (ALS). The authors showed that C9orf72 HRE impaired proximal axonal organelle motility, along with the DNA strand breaks and apoptosis. Interestingly, they claimed both gain and loss of functions contribute to C9orf72-linked ALS, and loss of function alone could induce axonal trafficking deficits. The authors

have conducted an interesting research and presented high quality data. Although there are some issues, this manuscript is overall promising and fully studied the distinct axonopathy driven by C9orf72 pathogenic HRE. Although the manuscript could be of sufficient interest to the readers of Life Sci Alliance, the following points need to be addressed by the authors:

1. The current manuscript suffers from the lack of mechanisms underlying C9orf72-linked axonal transport defects. For example, are the axonal transport of mitochondria and lysosome impaired through the similar mechanism?
2. For the C9orf72 GOF models, the authors should clarify the contribution of the C9orf72 DPRs and C9orf72 RNA repeats to the axonopathy, since increasing evidences in this field suggest that the PR and GR (and possibly GA) DPRs are more toxic than the other DPRs and the RNA repeats.
3. The authors simply combined the evidences of axonal trafficking deficits and DSB together, and the Reviewer wonders that is there any possible link between these two phenomena? If so, what is the potential mechanism?
4. The manuscript is overall well written but there are some writing issues. For example, Page 15: "to reveal the prominent DPR variants poly glycine-proline (GP) and poly glycine-alanine", there should be "GA" after glycine-alanine.

Reviewer #3 (Comments to the Authors (Required)):

In this manuscript, the authors use high content profiling to examine axonal transport in c9 repeat expansion derived motor neurons (c9-HRE), in c9 repeat expansion corrected motor neurons (c9-GC) and with knock out of the c9 gene (c9-KO), compared with WT as well. The authors show interesting data that c9-HRE lines look like Wt for distal and proximal neurite transport at D21 (looking at dye markers mitotracker and lysotracker), and but that WT lose transport distally ~D50-D80 and c9-HRE lose transport distally and proximally by D50-D80. The authors also KO c9 homozygously in WT or c9-HRE show defects in transport from even earlier. These are interesting data, although there are a number of issues that need to be addressed.

Major comments:

The authors here overstate their case regarding the role of the c9 gene because they are knocking out c9 homozygously, but in disease, it will be heterozygous. Therefore it is unclear if the c9 homozygous effects will be seen in a heterozygous situation. This needs to be brought up as a caveat of their findings, and other discussion will need to be toned down to account for this.

It is unclear how many cells are being examined in any one experiment, and if the experiments were performed blindly to the genotypes of the cultures. This could have an effect on bias.

In figure 1a, they should indicate the average distance of the neurites from the cell bodies, and the distance between proximal and distal measurements.

In all graphs, they should indicate individual data points (Figures 5 and 6), and there are no stats in figure 4. Is this because the results were not repeated?

Other comments:

It would be helpful to include a set of abbreviations, as they use a lot of abbreviations.

HC is not defined in the abstract.

They use incomplete language in a number of situations - as one example, deficient trafficking in aged C9orf72 on page 6 of the pdf. Presumably they mean deficient trafficking in aged motor neurons of c9orf72 patient cells?

What are all the cells in their cultures? They seem to ignore the possibility that non-autonomous interactions with for example glia may occur in vivo that may mean their in vitro data may not mimic in vivo.

On page 12 of the manuscript they indicate an isogenic set of additional MN models. It is really nice to have isogenic lines for a control, but they have only one. The way they state it sounds like they generated isogenic lines for every c9 disease line in hand.

How do they know that their c9-KO doesn't have an off target hit as well.

They refer to a pending report (under review at Science translational medicine). This language doesn't seem appropriate and they did not provide this ms to reviewers.

In the methods, they refer to many published approaches "as described". It would be appropriate to add some details (In brief...) for these.

Department of Neurology
at the Centre for Clinical Neurosciences
Gehlsheimer Straße 20 · 18147 Rostock
Germany

Director
Prof. Alexander Storch, MD

**Translational Neurodegeneration
Section „Albrecht Kossel“**

Head
Prof. Andreas Hermann, MD, PhD

Research Group
Translational Neurodegeneration
Group Leader: Prof. A. Hermann, MD, PhD
Phone: +49 381 494-9541
Fax: +49 381 494-9542
Andreas.Hermann@med.uni-rostock.de

Homepage
<https://albrecht-kossel-institut.med.uni-rostock.de/>

The Hermann and Lilly Schilling-Stiftung
within the Deutsche Stiftungszentrum
supports the Translational
Neurodegeneration Section with a 8-year
development award. More Information on
website of the Deutsches Stiftungszentrum:
www.deutsches-stiftungszentrum.de.

**HERMANN UND LILLY
SCHILLING-STIFTUNG
FÜR MEDIZINISCHE
FORSCHUNG**

11.12.2020

Point-by-point reply - LSA-2020-00764-T

Dear Reilly,

Please find below our point-by-point-reply. We deeply thank the reviewers for the very constructive comments. We tried our best to address all of them to the best of our knowledge, which really helped to improve our manuscript. By doing so, we believe that the manuscript now merits publication for the broad readership of Life Science Alliance.

No comments by Reviewer #1

Reviewer #2:

General remark:

In this manuscript, Pal et al. summarized a potential mechanism by which the C9orf72 hexanucleotide GGGGCC repeat expansion (HRE) induces amyotrophic lateral sclerosis (ALS). The authors showed that C9orf72 HRE impaired proximal axonal organelle motility, along with the DNA strand breaks and apoptosis. Interestingly, they claimed both gain and loss of functions contribute to C9orf72-linked ALS, and loss of function alone could induce axonal trafficking deficits. The authors have conducted an interesting research and presented high quality data.

Response: We deeply appreciate this enthusiastic review!

In cooperation with

1. *The current manuscript suffers from the lack of mechanisms underlying C9orf72-linked axonal transport defects. For example, are the axonal transport of mitochondria and lysosome impaired through the similar mechanism?*

Response: We recognize multiparametric HC profiling bears the potential to gain a systems view and to predict underlying pathways and candidate players. However, unlike in HT compound or genetic screens, we are currently lacking a sufficiently large database for matching our few phenotypic HC profiles for such comprehensive mechanistic predictions and subsequent hit validation experiments. Therefore, the purpose of our work was to reveal how HRE-mediated pathology in C9ORF compared phenotypically against mutant FUS and TDP43 (two other ALS-related mutations) and the different HC profiles indicate some clear differences in the underlying mechanisms. Mechanistic dissection down to molecular players and pathways in our opinion is beyond the scope of this work due to the lack of a large profile collection, database, etc. but we acknowledge there are some data available accompanying our HC profiles that hint to some underlying mechanisms which we show or refer to in an extended results and discussion section of the revised manuscript (e.g. completely new Figure 4). Our C9ORF lines were recently generated and characterized by Abo-Rady et al. (2020)¹. The authors performed transcriptomics by deep sequencing and as highlighted in figure 3 in that publication¹ the obtained GO terms point to many alterations of microtubules and motors, DNA damage response and apoptosis but not to anything specific for mitochondria and/or lysosomes.

We now further refined the bioinformatical analysis of these RNA Seq data¹ to further illuminate on systemic versus organelle-specific mechanisms in the revised version of our manuscript (new Figure 4). Specifically, we performed further protein-protein interaction (PPI) network mapping to reveal functional clusters in STRING. Also this in depth analysis of RNA Seq datasets clearly points towards general axonal transport problems rather than organelle specific things (“microtubule network”, “oxphos” etc.). This provides further hints that the trafficking defects as seen by us are likely due to a more upstream, presumably systemic, cause that impacts accordingly on different organelle types in a similar fashion. In line with this view is the gross similarity of the lysosomal versus mitochondrial parts in our HC profiles in Figure 1 and 3, especially towards the endpoints. Consistently, when we systemically disrupted all microtubule-dependent organelle trafficking with Nocodazole (our publication Pal et al., 2018)² our HC profiling revealed very similar alterations in both the mitochondrial and lysosomal parts as well (figure 2 in Pal et al., 2018)². Ditto when we inhibited mitochondrial ATP production with Oligomycin A as the resultant energy deprivation systemically reduced axonal trafficking of all organelle types investigated (figure 2 in Pal et al., 2018)². We added these important arguments to the revised Results (new section ‘Axonal trafficking defects in ageing C9ORF72 MNs were due to common, systemic perturbations of the cytoskeleton and energy supply’) and Discussion. If the reviewer and editor felt it would help the reader of the current manuscript, we could ask for permission to reprint these results which are already published elsewhere² also in this paper.

2. For the *C9orf72* GOF models, the authors should clarify the contribution of the *C9orf72* DPRs and *C9orf72* RNA repeats to the axonopathy, since increasing evidences in this field suggest that the PR and GR (and possibly GA) DPRs are more toxic than the other DPRs and the RNA repeats.

RNA foci in different time points

Response: We understand that the reviewer is asking for a further dissection of the GOF mechanisms to evaluate the contribution of RNA repeat expansions (RREs, causing toxic transcriptional RNAi foci) versus DPRs which are RAN-translated from these RRE transcripts. We appreciate such further dissection would indeed add more beneficial mechanistic insights. However, specifically addressing RREs against DPRs is technically very challenging and seemingly not fully achieved even by leading experts in the field. Addressing RRE-mediated RNA-foci requires expressing RRE constructs without translating DPRs. Such expression plasmids (also established as adenoviral vectors) were established by few investigators^{3,4}. These plasmids are standard mammalian expression constructs such as pcDNA5⁴ and have all start codons in the expression cassette removed to enable RRE transcription from G4C2 hexanucleotide repeats with no DPR translation. Episomal, transient expression leads indeed to increased RRE foci detectable through FISH probes. However, we are not aware of any publication that has ever disproven that these plasmid-derived RRE transcripts are not RAN-translated as well, thereby again leading to co-translation of DPRs. There is no good reason to believe that the endogenous host RAN transcription machinery will strongly prefer only endogenous RRE transcripts whilst leaving episomal plasmid-derived RREs untranslated. On the contrary, Scoles et al. (2015)⁵ have clearly demonstrated RAN translation from plasmids with CMV promoters. Further to the vice versa expression of DPRs without RREs, both Walker et al. (2017)⁴ and Callister et al. (2016)³ have established codon-optimized DPR expression plasmids that avoid the homogenous G4C2 repetitiveness by exploiting the degenerated genetic code to obtain a more heterogeneous codon composition. However, both investigators have failed to disprove that these codon-optimized plasmids do not transcribe RRE-foci. On the contrary, Walker et al. (2017)⁴ clearly documented the occurrence of R-loops upon expression of their DPR plasmids. R-loops normally arise from augmented RRE-transcripts in RNA foci, therefore one cannot rule out the possibility that these codon-optimized DPR plasmids retain a sufficient degree of repetitiveness and C-G-interactions for RRE-mediated toxicity. Finally, as iPSC-derived cell lines are knowingly very difficult to transfect, we would have to establish lentiviral tools to express these RRE and DPR constructs in our cellular model or otherwise generate stable lines using CRISP/Cas9 gene editing technology. Both approaches are clearly beyond the scope and time line of a reasonable revision, especially because after a period of forgoing technical development we still would have to add the time requirement for our extended time course experiments (until D80 and at least three experiments). Ditto for the microinjection of recombinant DPRs into neurons in culture that would require many preliminary tests, validations and optimizations as this approach is technically very challenging and novel and has not been established by anyone so far.

Collectively, we disfavor the expression of these RRE and DPR constructs as they do not separate RRE transcription sufficiently from DPR translation. Moreover, the strength of our experimental strategy so far was to investigate HRE-mediated GOF versus LOF of C9ORF in the parental and edited endogenous

gene locus only. Adding episomal or viral expression will compromise our conceptually sound system as the obtained overexpression artifacts are difficult to interpret. Overexpressed RREs and DPRs, in addition to the questionable power to separate both effects, are each likely to have some detrimental impact anyway.

Therefore, we wish to point out that the main goal of our study was to establish the HRE-mediated GOF versus the LOF of C9ORF, which we have clearly achieved with our isogenic, endogenous GC and KO lines. How RRE-mediated GOF compares against DPR-mediated GOF is a somewhat different question and technically very difficult to address, thereby calling for a whole, separate project rather than revision and as mentioned above yet not successfully done to the best of our knowledge.

Independent of that we want to point out that also the WZ-KO line showed increased DNA damage and axonopathy, thus the main driver for those could be rather the C9ORF72 loss of function and the GOF mechanisms maybe only amplifiers. This is also underpinned by the data presented by Abo-Rady et al. (2020)¹ in which DPR were reduced in C9-KO lines compared to C9 lines (still higher than in C9-GC).

Nevertheless, we performed several new correlative time course studies to find out how alterations in marker levels of RNA toxicity (RNA foci and R-loops) compared against DPR in occurring either before or after the onset of the axonal trafficking defects as shown in figure 3. Specifically, we did FISH (for RNA foci), IF stainings for R-loops and GP over a time course experiments to assess DPR- versus RRE-mediated toxicity. As shown in new figure 8, a sudden increase of GP foci concurred with the onset of axonal trafficking defects at D40. Moreover, augmented RRE foci (=RNA foci) were present already at the beginning of our time courses in all C9ORF72 lines and steadily further increasing over time towards D40. This clearly suggests that the buildup of DPR deposits is the actual driving force behind the axonal perturbations (and the later DNA damage accumulation, see below), possibly in concert with RNA foci. Conversely, there wasn't any alteration in R-loops. Interestingly DNA damage became apparent far later than axon trafficking deficits (new fig. 8a) and SSB and DSB induction did not phenocopy C9ORF72 (new fig. S9).

3. The authors simply combined the evidences of axonal trafficking deficits and DSB together, and the Reviewer wonders that is there any possible link between these two phenomena? If so, what is the potential mechanism?

Response: This is a very interesting question but we also feel that deciphering this mechanistic link is beyond the scope of a revision. Nevertheless, we did novel correlative and mimic studies which provided new mechanistic insights.

In Naumann et al. (2018)⁶ our group has revealed a feasible link between impaired DNA damage response and deficient distal axonal organelle trafficking via a postulated nucleo-axonal crosstalk in mutant FUS. Similarly, we used inducers of DNA damage (different mode of action, e.g. TOP1, TOP2 inhibitors) in healthy controls and performed HC imaging profiling with the overall question whether DNA damage induction and if which one is sufficient to induce a C9ORF72 like HC profile. We thus used topoisomerase 1 poisons camptothecin (Top1, inducing SSBs) and etoposide (Top2, inducing DSBs) together with the Top2 inhibitor BNS-22 known not to induce DNA damage. While DSB induction caused HC profile mimicking FUS-ALS phenotypes (new figure S9), neither SSB

induction/Top1 poisoning nor Top2 inhibition led to a phenocopy of C9ORF72. Thus, axonal trafficking defects in C9ORF occur independently and prior to DNA damage (as further verified in our new correlative time course analysis in figure 8a) and DPR accumulation indicating DNA damage occurs rather downstream of axonal transport deficiency. We propose in revised Results and Discussion that DSBs arise as consequence from DPR accumulation, consistent with published data⁴. Unfortunately, we technically failed investigating oxidative DNA damage (by 8OhdG) in iPSC-derived motoneurons due to the poor commercial antibody.

Finally, we describe for the first time a somehow surprisingly strong phenotype in wild type C9ORF72 knock out neurons (WT-KO), which more resembled an augmented distal to proximal dye-back phenotype. Of note, DNA damage was also highly present while cell death was lower than in C9 mutants. We hypothesize that these phenomenon are different than the ones in the C9 mutant lines and may be through Rho GTPases and actin remodeling (see figures 7b, 8a and Discussion).

Thus, in summary, we believe that we provide now significant more mechanistic insights in the pathophysiology of C9ORF72 ALS.

4. *The manuscript is overall well written but there are some writing issues. For example, Page 15: "to reveal the prominent DPR variants poly glycine-proline (GP) and poly glycine-alanine", there should be "GA" after glycine-alanine.*

Response: We appreciate this positive statement and did carefully proofread our revised manuscript to eliminate such mistakes.

Reviewer #3:

General remark:

In this manuscript, the authors use high content profiling to examine axonal transport in c9 repeat expansion derived motor neurons (c9-HRE), in c9 repeat expansion corrected motor neurons (c9-GC) and with knock out of the c9 gene (c9-KO), compared with WT as well. The authors show interesting data that c9-HRE lines look like Wt for distal and proximal neurite transport at D21 (looking at the markers mitotracker and lysotracker), and but that WT lose transport distally ~D50-D80 and c9-HRE lose transport distally and proximally by D50-D80. The authors also KO c9 homozygously in WT or c9-HRE show defects in transport from even earlier. These are interesting data.

Response: We thank the reviewer for this overall positive statement

Major comments:

1. *The authors here overstate their case regarding the role of the c9 gene because they are knocking out c9 homozygously, but in disease, it will be heterozygous. Therefore it is unclear if the c9 homozygous effects will be seen in a heterozygous situation. This needs to be brought up as a caveat of their findings, and other discussion will need to be toned down to account for this.*

Response: We agree with the reviewer and highlight this point appropriately in the revised Discussion of the manuscript. Specifically, we appreciate the reviewer has correctly recognized that the CRISPR/Cas9n-mediated excision of the start codon in exon 2 of the C9ORF72 gene locus has led to a deactivated translation of both alleles with no remaining C9ORF72 expression (figure 1 in Abo-Rady, 2020)¹. This strategy, unlike conventional KOs, enabled to eliminate C9ORF72 translation without compromising RAN translation from intron 1. Technically, this strategy is unsuitable to target only one allele, therefore heterozygous lines are hard, if not impossible, to obtain. Conceptually, we disfavoured a heterozygous line because the purpose of our KO lines was not to clinically mimic the LOF of C9ORF72 by carefully fine-tuning its levels moderately down to the levels seen in clinical patients. Rather, we wished to clarify whether there is a LOF contribution to the overall pathology *per se*. Therefore, we sought to unmask the LOF in the complex, potentially obscuring, interplay with other multiple GOF mechanisms at maximum clarity. A heterozygous KO line bears the risk of a compensatory upregulation of the remaining functional allele, thereby compromising the clarity and conclusiveness of the data. A fine-tuned downregulation of C9ORF72 to find out how much reduction is required to evoke typical hallmarks of ALS is a somewhat different biological question and is better achieved with adjustable (i.e. inducible) KD vectors in a separate project. However, the reviewer is of course right in pointing out that our findings with homozygous KO lines do not faithfully mirror the clinic of ALS. We illuminate now on the technicalities, purpose and aim of the KO lines in more detail in the revised manuscript and have carefully pointed out the limits, mechanistic and clinical relevance of our findings in the Discussion accordingly.

2. It is unclear how many cells are being examined in any one experiment, and if the experiments were performed blindly to the genotypes of the cultures. This could have an effect on bias.

Response: We apologize these important points were not comprehensively enough addressed under Material and Methods. We have now revised this section and added a new one 'Bulk statistics, replicas and blinding' that provides further details. In essence, the number of imaged axons and obtained organelle tracks was very high and blinding was achieved on several levels.

3. In figure 1a, they should indicate the average distance of the neurites from the cell bodies, and the distance between proximal and distal measurements.

Response: We have included these dimensions now in revised figure 1a. According to the manufacture's (Xona Microfluidics) specifications, the distance between distal and proximal measurements in the microchannels is 900 μm . Concerning the average distance of the neurite (i.e. the microchannel entry) from the soma, we took the half width of the flanking proximal chamber (i.e. $2\text{mm}/2=1\text{mm}$) as the average distance of question, assuming that the neurons were homogeneously seeded here.

4. In all graphs, they should indicate individual data points (Figures 5 and 6), and there are no stats in figure 4. Is this because the results were not repeated?

Response: We changed all bar graphs to scatter plot format to better show the individual data points. Formally figure 4 (which is now supplemental figure S3 & S4) is based on the sum of z-scores. Z-scores

themselves are based on the standard deviation to better visualize differences between two datasets. A more detailed account is now provided in the last paragraph of the revised Result section 'Both gain and loss of function contribute to trafficking deficiency in C9ORF72'.

Minor comments:

1. *It would be helpful to include a set of abbreviations, as they use a lot of abbreviations.*

Response: We included a list of abbreviations in the revised manuscript (new table 2).

2. *HC is not defined in the abstract.*

Response: We now defined HC in the abstract.

3. *They use incomplete language in a number of situations - as one example, deficient trafficking in aged C9orf72 on page 6 of the pdf. Presumably they mean deficient trafficking in aged motor neurons of c9orf72 patient cells?*

Response: We apologized for these mistakes and thoroughly revised the manuscript. As a general remark, we are using the term 'in aged C9ORF72 MNs' or alike quite often throughout the manuscript, particularly in subheadings. Since we explain our disease model in the Introduction and did not employ any other model system, it should be clear that our streamlined term above always refers to iPSC-derived spinal motoneurons from patients during late maturation, i.e. ageing. We would prefer to avoid too much redundancy by using the full, long term over and over again. Misunderstandings for readers are unlikely because other model systems are only mentioned in a focused passage within the Introduction about third party preliminary work.

4. *What are all the cells in their cultures? They seem to ignore the possibility that non-autonomous interactions with for example glia may occur in vivo that may mean their in vitro data may not mimic in vivo.*

Response: Please refer to our previously published datasets and the one of others on the respective differentiation protocol. Using this protocols, we achieve nearly pure neuronal cultures, and all remaining non neuronal cells are left over progenitor cells. The protocol does not induce glial cells at all (PMID: 23533608, PMID: 30422121, PMID: 32084385, PMID: 29362359). Moreover, we illuminate on the purity of our MN cultures at the end of the new section 'Final maturation and ageing of MNs in MFCs' in revised Material and Methods. In essence, in the proximal seeding chamber, our differentiation protocol yields up to 80% mature motoneurons and a remaining mix of neuronal progenitors and other neuronal subtypes. The cultures are devoid of glia cells and astrocytes. Neurites penetrating the microchannels and sprouting out at the distal exit are virtually 100% axon-pure (see Glass et al., 2020)⁷. Therefore, our culture system is indeed unsuitable to model non-autonomous interactions. But the scope of our manuscript was to investigate endogenous HRE-mediated GOF and LOF mechanisms in motoneurons. This is why we chose our model system. The reviewer seems to raise a general awareness of the pros, cons and limits of each model system which we in principle acknowledge and appreciate. But obviously, disease modelling on composite cultures is not the aim of our study.

5. On page 12 of the manuscript they indicate an isogenic set of additional MN models. It is really nice to have isogenic lines for a control, but they have only one. The way they state it sounds like they generated isogenic lines for every c9 disease line in hand.

Response: We apologize for the unclear phrase and have revised the sentence of question in Results, section 'Both gain and loss of function contribute to trafficking deficiency in C9ORF72'. Moreover, please refer to table 1. To have a better overview we highlighted isogenic groups in blue (revised table 1). As shown there, we have an isogenic pair for the comparison WT (Ctrl1) vs. WT-KO and additionally an isogenic trio for C9 vs. C9-GC and C9-KO analyzed in the axon trafficking/DNA damage studies etc.

6. How do they know that their C9-KO doesn't have an off target hit as well.

Response: Given the advanced target specificity of latest generation CRISPR/Cas9n tools, this possibility appears very unlikely. Moreover, Abo-Rady et al. (2020)¹ have characterized two C9-KO lines which were phenotypically indistinguishable, thereby further arguing against off target effects.

7. They refer to a pending report (under review at Science translational medicine). This language doesn't seem appropriate and they did not provide this ms to reviewers.

Response: We apologize for this misunderstanding. As correctly cited, this paper is openly available at BioRxiv, please refer to <https://www.biorxiv.org/content/10.1101/835082v3>.

8. In the methods, they refer to many published approaches "as described". It would be appropriate to add some details (In brief...) for these.

Response: We appreciate this comment and included brief further descriptions wherever appropriate. New details were added predominantly to the revised Material and Methods section 'Generation, gene-editing and differentiation of human iPSC cell lines to MNs' and the new section 'Final maturation and ageing of MNs in MFCs'.

Yours sincerely,

Andreas Hermann
Corresponding author

References

- 1 Abo-Rady, M. *et al.* Knocking out C9ORF72 Exacerbates Axonal Trafficking Defects Associated with Hexanucleotide Repeat Expansion and Reduces Levels of Heat Shock Proteins. *Stem Cell Reports* **14**, 390-405, doi:10.1016/j.stemcr.2020.01.010 (2020).
- 2 Pal, A. *et al.* High content organelle trafficking enables disease state profiling as powerful tool for disease modelling. *Scientific data* **5**, 180241, doi:10.1038/sdata.2018.241 (2018).
- 3 Bennion Callister, J., Ryan, S., Sim, J., Rollinson, S. & Pickering-Brown, S. M. Modelling C9orf72 dipeptide repeat proteins of a physiologically relevant size. *Human molecular genetics* **25**, 5069-5082, doi:10.1093/hmg/ddw327 (2016).
- 4 Walker, C. *et al.* C9orf72 expansion disrupts ATM-mediated chromosomal break repair. *Nature neuroscience* **20**, 1225-1235, doi:10.1038/nn.4604 (2017).
- 5 Scoles, D. R. *et al.* Repeat Associated Non-AUG Translation (RAN Translation) Dependent on Sequence Downstream of the ATXN2 CAG Repeat. *PloS one* **10**, e0128769, doi:10.1371/journal.pone.0128769 (2015).
- 6 Naumann, M. *et al.* Impaired DNA damage response signaling by FUS-NLS mutations leads to neurodegeneration and FUS aggregate formation. *Nat Commun* **9**, 335, doi:10.1038/s41467-017-02299-1 (2018).

January 19, 2021

RE: Life Science Alliance Manuscript #LSA-2020-00764-TR

Prof. Andreas Hermann
University of Rostock
Gehlsheimer Str. 20
Rostock 18147
Germany

Dear Dr. Hermann,

Thank you for submitting your revised manuscript entitled "Concomitant gain and loss of function pathomechanisms in C9ORF72 amyotrophic lateral sclerosis". We would be happy to publish your paper in Life Science Alliance pending final revisions necessary to meet our formatting guidelines.

Along with the points listed below, please also attend to the following:

- please consult our manuscript preparation guidelines <https://www.life-science-alliance.org/manuscript-prep> and make sure your manuscript sections are in the correct order
- please deposit the protein-protein interaction data in a publicly available database and include the accession number in the revised manuscript
- please make sure the author order in your manuscript and our system match
- please add your supplementary figure & video legends to the main manuscript text, after the main figure legends
- please double-check your figure callouts: please add callouts for Figures 5B, 7A, and S11 to your main manuscript text; there are callouts for Figure S6 A,B,C but there is no mention of panels A,B,C in the actual Figure or Figure Legend; there is a typo: Figure 54A on page 72
- please make sure that the insets in Fig 6A Control panels 1 and 2 match the zoomed in images
- please use the [10 author names, et al.] format in your references (i.e. limit the author names to the first 10)
- please add the Author Contributions of all Authors to your main manuscript text
- please upload your supplementary figures as single files

A. FINAL FILES:

B. MANUSCRIPT ORGANIZATION AND FORMATTING:

Sincerely,

Shachi Bhatt, Ph.D.

Executive Editor
Life Science Alliance
<https://www.lsjournal.org/>
Tweet @SciBhatt @LSAJournal

Reviewer #2 (Comments to the Authors (Required)):

The authors have addressed the comments and concerns raised by the Reviewer. Now the revised manuscript is suitable for publication.

Reviewer #3 (Comments to the Authors (Required)):

The authors have addressed my concerns. This is an interesting paper trying to sort out gain of function from loss of function effects of the c9orf72 mutation on the function of the c9orf72 gene.

January 27, 2021

RE: Life Science Alliance Manuscript #LSA-2020-00764-TRR

Prof. Andreas Hermann
University of Rostock
Gehlsheimer Str. 20
Rostock 18147
Germany

Dear Dr. Hermann,

Thank you for submitting your Research Article entitled "Concomitant gain and loss of function pathomechanisms in C9ORF72 amyotrophic lateral sclerosis". It is a pleasure to let you know that your manuscript is now accepted for publication in Life Science Alliance. Congratulations on this interesting work.

DISTRIBUTION OF MATERIALS:

Again, congratulations on a very nice paper. I hope you found the review process to be constructive and are pleased with how the manuscript was handled editorially. We look forward to future exciting submissions from your lab.

Sincerely,

Shachi Bhatt, Ph.D.

Executive Editor

Life Science Alliance

<https://www.lsjournal.org/>
